# Constrained Bayesian Optimization with Adaptive Active Learning of Unknown Constraints

## Abstract

Optimizing objectives under constraints, where both the objectives and constraints are black box functions, is a common scenario in real-world applications such as the design of medical therapies, industrial process optimization, and hyperparameter optimization. One popular approach to handling these complex scenarios is Bayesian Optimization (BO). In terms of theoretical behavior, BO is relatively well understood in the unconstrained setting, where its principles have been well explored and validated. However, when it comes to constrained Bayesian optimization (CBO), the existing framework often relies on heuristics or approximations without the same level of theoretical guarantees. In this paper, we delve into the theoretical and practical aspects of constrained Bayesian optimization, where the objective and constraints can be independently evaluated and are subject to noise. By recognizing that both the objective and constraints can help identify high-confidence *regions of interest* (ROI), we propose an efficient CBO framework that intersects the ROIs identified from each aspect to determine the general ROI. The ROI, coupled with a novel acquisition function that adaptively balances the optimization of the objective and the identification of feasible regions, enables us to derive rigorous theoretical guarantees for its performance. We showcase the efficiency and robustness of our proposed CBO framework through extensive empirical evidence and discuss the fundamental challenge of deriving practical regret bounds for CBO algorithms.

## 1 Introduction

Bayesian optimization (BO) has been widely studied as a powerful framework for expensive black-box optimization tasks in machine learning, engineering, and science in the past decades. Additionally, many real-world applications often involve black-box constraints that are costly to evaluate. Examples include choosing from a plethora of untested medical therapies under safety constraints (Sui et al., 2015); determining optimal pumping rates in hydrology to minimize operational costs under constraints on plume boundaries (Gramacy et al., 2016); or tuning hyperparameters of a neural network under memory constraints (Gelbart et al., 2014). To incorporate the constraints into the BO framework, it is common to model constraints analogously to the objectives via Gaussian processes (GP) and then utilize an acquisition function to trade off the learning and optimization to decide subsequent query points.

Over the past decade, advancements have been made in several directions trying to address constrained BO (CBO). However, limitations remain in each direction. For instance, extended Expected Improvement (Gelbart et al., 2014; Gardner et al., 2014) approaches learn the constraints passively and can lead to inferior performance where the feasibility matters more than the objective optimization. The augmented lagrangian (AL) methods (Gramacy et al., 2016; Picheny et al., 2016; Ariafar et al., 2019) convert constrained optimization into unconstrained optimization with additional hyperparameters that are decisive for its performance while lacking theoretical guidance on the choice of its value. The entropy-based methods (Takeno et al., 2022) rely heavily on approximation and sampling, which violates the theoretical soundness of the method. In general, the current methods extend the unconstrained BO methods with approximation or heuristics to learn the constraints and optimize the objective simultaneously, lacking a rigorous performance guarantee as in the Bayesian optimization tasks without the need to learn unknown constraints.

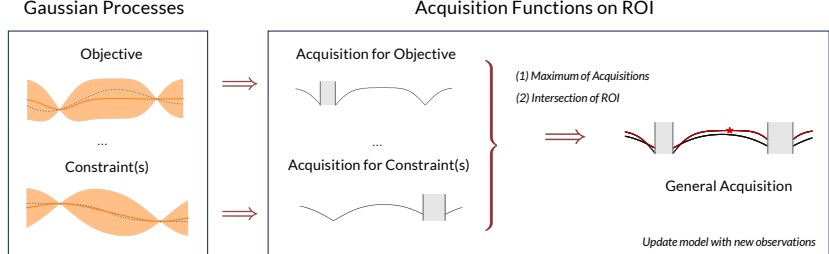

Figure 1: Pipeline of proposed algorithm COBALT. In the left box, we maintain a Gaussian process as the surrogate model for the unknown objective and each constraint. The dotted curve shows the actual function, the red curve shows the predicted mean, and the shaded area denotes the confidence interval. In the right box, we first derive the acquisitions from each Gaussian process defined on a corresponding region of interests and define the general acquisition function by combining them all. Each time, the algorithm maintains the model, maximizes the general acquisition function to pick the candidate to evaluate, and then updates the model with the new observation. In later sections, we will elaborate on the filtered gray gap in the acquisition.

The challenge and necessity of learning the unknown constraints for constrained BO motivate us to resort to active learning methods dealing with unknown constraints. Such methods have been studied under active learning for level-set estimation (AL-LSE). Much like BO, AL-LSE models the black-box function through a Gaussian process and pursues optimization via sequential queries. However, the distinction lies in the objectives: while BO focuses on finding the maximizer of an objective, AL-LSE seeks to classify points in the domain as lying above or below a specified threshold. This setting is particularly relevant in applications where a desirable region, rather than a single optimal point, is sought, as seen in environmental monitoring and sensor networks (Gotovos et al., 2013). Recent approaches, such as truncated variance reduction (Bogunovic et al., 2016) and an information-theoretic framework (Nguyen et al., 2021), aim to unify the theories of both sides. The unification inspires us to design a framework that adaptively balances the active feasible region identification and the unknown objective optimization.

In this paper, we propose a novel framework that integrates AL-LSE with BO for constrained Bayesian optimization. Our approach leverages the theoretical advantages of both paradigms, allowing for a rigorous performance analysis of the CBO method. A brief illustration of the framework design is shown in figure 1. The subsequent sections of this paper are structured as follows. In section 2, we provide a detailed overview of recent advancements in various facets of CBO. In section 3, we delve into the problem statement and discuss the definition of a probabilistic regret as a performance metric that enables rigorous performance analysis. In sections 4 and 5, we propose the novel CBO framework and offer the corresponding performance analysis. In section 6, we provide empirical evidence for the efficacy of the proposed algorithm. In section 7, we reflect on the key takeaways of our framework and discuss its potential implications for future work.

## 2    RELATED WORK

**Feasibility-calibrated unconstrained methods.**    While the majority of the research in Bayesian optimization (BO) is concerned with unconstrained problems as summarized by Frazier (2018), there exists works that also consider black-box constraints. The pioneering work by Schonlau et al. (1998) first extended Expected Improvement (EI) to constrained cases by defining at a certain point the product of the expected improvement and the probability of the point being feasible. Later, this cEI algorithm was further advanced by Gelbart et al. (2014) and Gardner et al. (2014), and the noisy setting was studied by Letham et al. (2019) using Monte Carlo integration while also introducing batch acquisition. The posterior sampling method (Thompson sampling) is extended to scalable CBO (SCBO) by Eriksson and Poloczek (2021). The proposed SCBO algorithm converts the original observation with a Gaussian copula. It addresses the heteroskedasticity of unknown functions over the large search space with axis-aligned trust region, extending the TuRBO (Eriksson et al., 2019) in the unconstrained BO with additional sampling from the posterior of the constrained functions to weight the samples of the objective. Some recent works (Zhou and Ji, 2022; Xu et al., 2023; Lu and Paulson, 2022) aim at different definitions of regret where the violation of constraints is either directly added to the regret as weighted penalization or studied separately from the regret incurred by the evaluation of objective function. They offer theoretical insights yet lack treatment for infinite regret when no reward is yielded when the evaluated points lie out of the feasible area. **In general, the problem with**

the feasibility-calibrated methods is the lack of guarantee that through optimization, there is a feasible point being picked. Workarounds, including maximizing the feasibility or minimizing the constraint violation, are introduced, damaging the soundness of the algorithm.

**Information-based criterion**   There has been a recent surge of interest in the field toward the information-theoretic framework within BO. Predictive entropy search (PES) (Hernández-Lobato et al., 2014) was extended to constraints (PESC) and detailed the entropy computations by Hernández-Lobato et al. (2015). Later, max-value entropy search was proposed by Wang and Jegelka (2017) to address the PES's expensive computations for estimating entropy, and the constrained adaptation was developed by Perrone et al. (2019). More recently, variants of the MES were developed, such as the methods based on a lower bound of mutual information (MI), which guarantees non-negativity. (Takeno et al., 2022) However, approximations, including sampling and variational inference, are introduced due to the intrinsic difficulty of the direct estimation of the entropy. Despite the strong empirical performance, the theoretical guarantee of these CBO extensions of entropy methods remains an open challenge.

**Additive-structure methods**   Another direction in this field incorporates BO into the augmented Lagrangian framework by placing the constraint into the objective function and reducing the problem into solving a series of unconstrained optimization tasks using vanilla BO (Gramacy et al., 2016). The idea was developed further by Picheny et al. (2016) using slack variables to convert mixed constraints to only equality constraints, which achieve better performance, and by Ariafar et al. (2019) using ADMM algorithm to tackle an augmented Lagrangian relaxation. Similarly, in the risk-averse BO setting studied by Makarova et al. (2021), the unknown heteroscedastic risk corresponding to the reward is considered a penalty. It is added with a manually specified coefficient. Their method comes with a rigorous theoretical guarantee regarding the corresponding risk-regulated reward. However, in both the risk-averse method and the augmented Lagrangian frameworks, the appropriate coefficient of the additive structure is essential to the performance while lacking a prior theoretical justification.

**Active learning of constraint(s)**   Though data-selection for active learning could date back to MacKay (1992), active learning for Level-set estimation (AL-LSE) was initially proposed by Gotovos et al. (2013) to perform a classification task on the sample space which enjoys theoretical guarantees. As both AL-LSE and BO share GP features, the method was later extended to the BO domain by Bogunovic et al. (2016), in which they unify the two under truncated variance reduction and assume the kernel such that the variance reduction function is submodular.  The problem with directly applying level-set estimation methods is their limitation on dealing with only one unknown function at one time and lack of a straightforward extension to trade-off the learning of multiple unknown functions, as typically seen in BO with unknown constraints. Malkomes et al. (2021); Komiyama et al. (2022) propose a novel acquisition function that prioritizes diversity in the active search. However, there is no straightforward extension to adaptively trade off the learning of constraints and the optimization of the objective. Incorporating its idea into our setting remains challenging.

## 3    PROBLEM STATEMENT

In this section, we introduce a few useful notations and formalize the problem. Consider a compact search space $\mathbf{X} \subseteq \mathbb{R}$. We aim to find a maximizer $x^* \in \arg\max_{\mathbf{x} \in \mathbf{X}} f(\mathbf{x})$ of a black-box function $f : \mathbf{X} \to \mathbb{R}$, subject to $K$ black-box constraints $\mathcal{C}_k(\mathbf{x})$ ($k \in \mathbf{K} = \{1, 2, 3, ..., K\}$) such that each constraint is satisfied by staying above its corresponding threshold $h_k$ [1]. Thus, formally, our goal can be formulated as finding:

$$\max_{\mathbf{x} \in \mathbf{X}} f(\mathbf{x}) \text{ s.t. } \mathcal{C}_k(\mathbf{x}) > h_k, \forall k \in \mathbf{K}$$

We maintain a Gaussian process ($\mathcal{GP}$) as the surrogate model for each black-box function, pick a point $\mathbf{x}_t \in \mathbf{X}$ at iteration $t$ by maximizing the acquisition function $\alpha : \mathbf{X} \to \mathbb{R}$, and observe the function values perturbed by additive noise: $y_{f,t} = f(\mathbf{x}_t) + \epsilon$ and $y_{\mathcal{C}_k,t} = \mathcal{C}_k(\mathbf{x}_t) + \epsilon$, with $\epsilon \sim \mathcal{N}(0, \sigma^2)$ being i.i.d. Gaussian noise. Each $\mathcal{GP}(m(\mathbf{x}), k(\mathbf{x}, \mathbf{x}'))$ is fully specified by its prior

---

[1]Note that the minimization problem and the case $\mathcal{C}_k(\mathbf{x}) < h_k$ are captured in this formalism, as $f(\mathbf{x})$ and $\mathcal{C}_k(\mathbf{x})$ can both be negated.

mean $m$ and kernel $k$. With the historical observations $\mathbf{D}_{t-1} = \{(\mathbf{x}_i, y_{f,i}, \{y_{c_K,i}\}_{k \in \mathbf{K}})\}_{i=1,2,\ldots t-1}$, the posterior also takes the form of a $\mathcal{GP}$, with mean

$$\mu_t(\mathbf{x}) = k_t(\mathbf{x})^\top (\mathbf{K}_t + \sigma^2 I)^{-1} \mathbf{y}_t \tag{1}$$

and covariance

$$k_t(\mathbf{x}, \mathbf{x}') = k(\mathbf{x}, \mathbf{x}') - k_t(\mathbf{x})^\top (\mathbf{K}_t + \sigma^2 I)^{-1} k_t(\mathbf{x}') \tag{2}$$

where $k_t(\mathbf{x}) \triangleq [k(\mathbf{x}_1, \mathbf{x}), \ldots, k(\mathbf{x}_t, \mathbf{x})]^\top$ and $\mathbf{K}_t \triangleq [k(\mathbf{x}, \mathbf{x}')]_{\mathbf{x}, \mathbf{x}' \in \mathbf{D}_{t-1}}$ is a positive definite kernel matrix (Rasmussen and Williams, 2006).

The definition of reward plays an important role in analyzing online learning algorithms. Throughout the rest of the paper, we define the reward of CBO as the following and defer the detailed discussion to Appendix C.

$$r(\mathbf{x}) = \begin{cases} f(\mathbf{x}) & \text{if } \mathbb{I}(C_k(\mathbf{x}) > h_k) \quad \forall k \in \mathbf{K} \\ -\inf & \text{o.w.} \end{cases} \tag{3}$$

For simplicity and without loss of generality, we stick to the definition in equation 3 and let all $h_k = 0$. We want to locate the global maximizer efficiently

$$\mathbf{x}^* = \underset{\mathbf{x} \in \mathbf{X}, \forall k \in \mathbf{K}, \mathcal{C}_k(\mathbf{x}) > 0}{\arg\max} f(\mathbf{x})$$

Equivalently, we seek to achieve the performance guarantee in terms of simple regret at certain time $t$,

$$\mathbf{R}_t := r(\mathbf{x}^*) - \underset{\mathbf{x} \in \{\mathbf{x}_1, \mathbf{x}_2, \ldots \mathbf{x}_t\}}{\max} r(\mathbf{x})$$

with a certain probability guarantee. Formally, given a certain confidence level $\delta$ and constant $\epsilon$, we want to guarantee that after using up certain budget $T$ dependent on $\delta$ and $\epsilon$, we could achieve a high probability upper bound of the simple regret on the identified area $\hat{\mathbf{X}}$ which is the subset of $\mathbf{X}$.

$$P(\underset{\mathbf{x} \in \hat{\mathbf{X}}}{\max} \mathbf{R}_T(\mathbf{x}) \geq \epsilon) \leq 1 - \delta$$

## 4 THE COBALT ALGORITHM

**We start with necessary concepts from the active learning for level-set estimation and delve into the framework design.**[2]

### 4.1 ACTIVE LEARNING FOR LEVEL-SET ESTIMATION

We follow the common practice and assume each unknown constraint or objective is sampled from a corresponding independent Gaussian process ($\mathcal{GP}$) (Hernández-Lobato et al., 2015; Gelbart et al., 2014; Gotovos et al., 2013) to treat the epistemic uncertainty.

$$\mathcal{C}_k \sim \mathcal{GP}_{\mathcal{C}_k} \quad \forall k \in \mathbf{K}$$
$$f \sim \mathcal{GP}_f$$

We could derive pointwise confidence interval estimation with the $\mathcal{GP}$ for each black-box function. We define the upper confidence bound $\mathrm{UCB}_t(\mathbf{x}) \triangleq \mu_{t-1}(\mathbf{x}) + \beta_t^{1/2} \sigma_{t-1}(\mathbf{x})$ and lower confidence bound $\mathrm{LCB}_t(\mathbf{x}) \triangleq \mu_{t-1}(\mathbf{x}) - \beta_t^{1/2} \sigma_{t-1}(\mathbf{x})$, where $\sigma_{t-1}(\mathbf{x}) = k_{t-1}(\mathbf{x}, \mathbf{x})^{1/2}$ and $\beta$ acts as a scaling factor corresponding to certain confidence. For each unknown constraint $\mathcal{C}_k$, we follow the notations from Gotovos et al. (2013) and define the superlevel-set to be the areas that meet the constraint $\mathcal{C}_k$ with high confidence

$$S_{\mathcal{C}_k,t} \triangleq \{\mathbf{x} \in \mathbf{X} | \mathrm{LCB}_{\mathcal{C}_k,t}(\mathbf{x}) > 0\}$$

We define the sublevel-set to be the areas that do not meet the constraint $\mathcal{C}_k$ with high confidence

$$L_{\mathcal{C}_k,t} \triangleq \{\mathbf{x} \in \mathbf{X} | \mathrm{UCB}_{\mathcal{C}_k,t}(\mathbf{x}) < 0\}$$

and the undecided set is defined as

$$U_{\mathcal{C}_k,t} \triangleq \{\mathbf{x} \in \mathbf{X} | \mathrm{UCB}_{\mathcal{C}_k,t}(\mathbf{x}) \geq 0, \mathrm{LCB}_{\mathcal{C}_k,t}(\mathbf{x}) \leq 0\}$$

where the points remain to be classified.

---

[2]There was another COBALT algorithm for CBO proposed by Paulson and Lu (2022).

## 4.2 REGION OF INTEREST IDENTIFICATION FOR EFFICIENT CBO

In the CBO setting, we only care about the superlevel-set $S_{\mathcal{C}_k,t}$ and undecided-set $U_{\mathcal{C}_k,t}$, where the global optimum is likely to lie in. Hence, we define the region of interest for each constraint function $\mathcal{C}_k$ as

$$\hat{\mathbf{X}}_{\mathcal{C}_k,t} \triangleq S_{\mathcal{C}_k,t} \cup U_{\mathcal{C}_k,t} = \{\mathbf{x} \in \mathbf{X} | \text{UCB}_{\mathcal{C}_k,t}(\mathbf{x}) \geq 0\}$$

Similarly, for the objective function, though there is no pre-specified threshold, we could use the maximum of $\text{LCB}_f(x)$ on the intersection of superlevel-set $S_{\mathcal{C},t} \triangleq \bigcap_k^{\mathbf{K}} S_{\mathcal{C}_k,t}$

$$\text{LCB}_{f,t,max} \triangleq \begin{cases} \max_{\mathbf{x} \in S_{\mathcal{C},t}} \text{LCB}_{f,t}(\mathbf{x}), & \text{if } S_{\mathcal{C},t} \neq \emptyset \\ -\infty, & \text{o.w.} \end{cases}$$

as the high confidence threshold for the $\text{UCB}_{f,t}(\mathbf{x})$ to identify a region of interest for the optimization of the objective. Given that $\text{UCB}_{f,t}(\mathbf{x}^*) \geq f^* \geq f(\mathbf{x}) \geq \text{LCB}_{f,t}(\mathbf{x})$ with the probability specified by the choice of $\beta_{f,t}$ and $\beta_{\mathcal{C},t}$, we define the ROI for the objective optimization as

$$\hat{\mathbf{X}}_{f,t} \triangleq \{\mathbf{x} \in \mathbf{X} | \text{UCB}_{f,t}(\mathbf{x}) \geq \text{LCB}_{f,t,max}\}$$

By taking the intersection of the ROI of each constraint, we could identify the ROI for identifying the feasible region

$$\hat{\mathbf{X}}_{\mathcal{C},t} \triangleq \bigcap_k^{\mathbf{K}} \hat{\mathbf{X}}_{\mathcal{C}_k,t}$$

The combined ROI for CBO is determined by intersecting the ROIs of constraints and the objective:

$$\hat{\mathbf{X}}_t \triangleq \hat{\mathbf{X}}_{f,t} \cap \hat{\mathbf{X}}_{\mathcal{C},t} \tag{4}$$

## 4.3 COMBINING ACQUISITION FUNCTIONS FOR CBO

**Acquisition function for optimizing the objective** To optimize the unknown objective $f$ when $\hat{\mathbf{X}}_t$ is established, we can employ the following acquisition function [3]

$$\alpha_{f,t}(\mathbf{x}) \triangleq \begin{cases} \mathbf{UCB}_{f,t}(\mathbf{x}) - \mathbf{LCB}_{f,t,max} & \mathbf{LCB}_{f,t,max} \neq -\infty \\ \mathbf{UCB}_{f,t}(\mathbf{x}) - \mathbf{LCB}_{f,t}(\mathbf{x}) & \mathbf{LCB}_{f,t,max} = -\infty \end{cases} \tag{5}$$

At given $t$, to efficiently optimize the black-box $f$ we evaluate the point $\mathbf{x}_t = \arg\max_{\mathbf{x} \in \hat{\mathbf{X}}_t} \alpha_{f,t}(\mathbf{x})$. Since at a given $t$, when $\mathbf{LCB}_{f,t,max}(\mathbf{x})$ is constant, the acquisition function is equivalent to $\mathbf{UCB}_{f,t}(\mathbf{x})$.

**Acquisition function for learning the constraints** When we merely focus on identifying the feasible region defined by a certain unknown constraint $\mathcal{C}_k$, we could apply the following active learning acquisition function that could be dated back to MacKay (1992).

$$\alpha_{\mathcal{C}_k,t}(\mathbf{x}) \triangleq \mathbf{UCB}_{\mathcal{C}_k,t}(\mathbf{x}) - \mathbf{LCB}_{\mathcal{C}_k,t}(\mathbf{x}) \tag{6}$$

At given $t$, we evaluate the point $\mathbf{x}_t = \arg\max_{\mathbf{x} \in U_{\mathcal{C}_k,t} \cap \hat{\mathbf{X}}_t} \alpha_{\mathcal{C}_k,t}(\mathbf{x})$ to efficiently identify the feasible region defined by $\mathcal{C}_k$. Note that the acquisition function $\alpha_{\mathcal{C}_k,t}(\mathbf{x})$ is not maximized on the full $\hat{\mathbf{X}}_{\mathcal{C}_k,t}$, but only on $U_{\mathcal{C}_k,t} \cap \hat{\mathbf{X}}_t$ as is shown in figure 2. The active learning on the superlevel-set $S_{\mathcal{C}_k,t} \cap \hat{\mathbf{X}}_t$ doesn't contribute to identifying the corresponding feasible region.

**The COBALT acquisition criterion** With the two acquisitions discussed above and the ROIs discussed in section 4.2, we propose the algorithm COnstrained BO with Adaptive active Learning of unknown constraints (COBALT). [4] COBALT essentially picks a data point with the maximum

---

[3]Such criterion has been studied under the unconstrained setting (Zhang et al., 2023).

[4]We briefly discuss the possible extension to decoupled setting, where the objective and constraints may be evaluated independently, of COBALT in Appendix B.

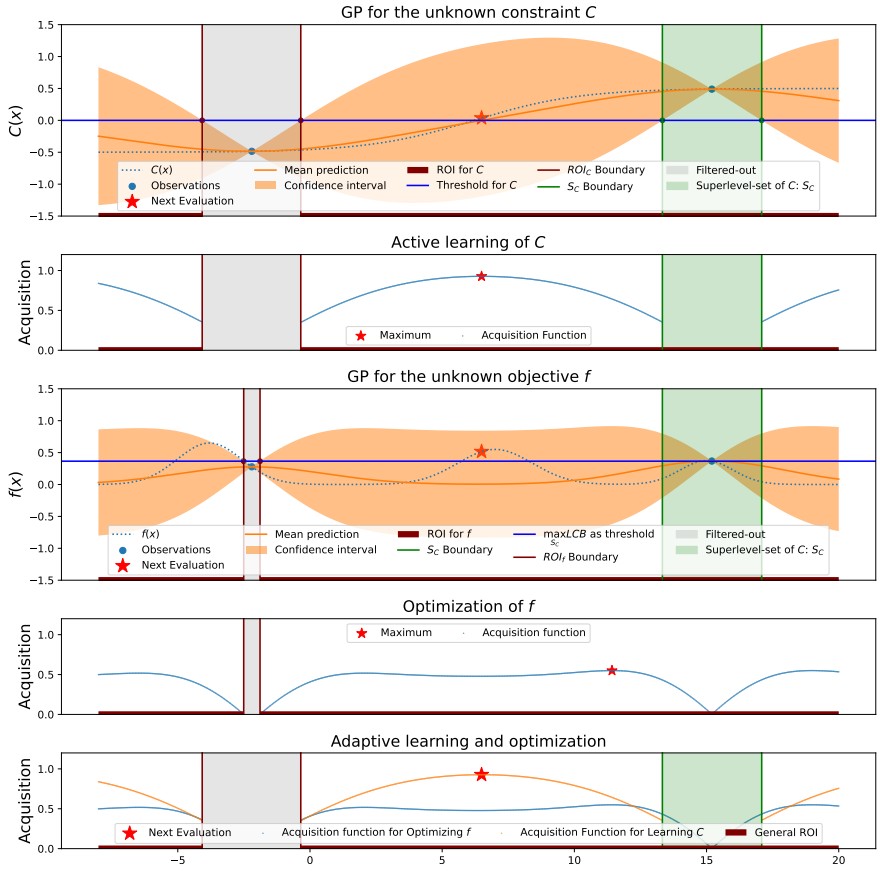

Figure 2: Illustration of COBALT on a synthetic noise-free 1D example. The first two rows show the GP for the $\mathcal{C}$, the superlevel-set $S_{\mathcal{C}}$, the region of interest $\hat{\mathbf{X}}_{\mathcal{C}}$ and the corresponding acquisition function $\alpha_{\mathcal{C}_k, t}(\mathbf{x})$ as defined in equation 6. The following two rows show the GP for $f$, the region of interest $\hat{\mathbf{X}}_f$, and the corresponding acquisition function $\alpha_{f,t}(\mathbf{x})$ defined in equation 5. We show that after identifying $S_{\mathcal{C}}$, we could define the threshold for ROI identification of $f$ accordingly. The bottom row demonstrates that the general ROI $\hat{\mathbf{X}}$ as defined in equation 4 is identified by taking the intersection ROI for $f$ and $\mathcal{C}$. The general acquisition function is defined as the maximum of the acquisition for $f$ and $\mathcal{C}$ and is maximized on the $\hat{\mathbf{X}}$. The scaling and length scale of the Gaussian processes are learned by maximizing the likelihood.

acquisition function value across all the acquisition functions defined on different domains. The maximization of different acquisition functions allows an adaptive tradeoff between the active learning of the constraints and the Bayesian Optimization of the objective on the feasible region. The intersection of ROIs allows an efficient search space shrinking for CBO. The complete procedure is shown in algorithm 1[5]. We also illustrate the procedure on a 1D toy example in figure 2. We construct the example to demonstrate that the explicit, active learning of the constraint doesn't necessarily hurt the optimization but could contribute directly to the simple regret improvement. **Note that in practice, the algorithm searches on a finite discretization $\tilde{D}$ of $\mathbf{X}$. The membership of each element to the ROIs could be checked in a pointwise fashion.**

## 5   THEORETICAL ANALYSIS

We first state a few assumptions that provide insights into the convergence properties of COBALT.

**Assumption 1** *The objective and constraints are sampled from independent Gaussian processes. Formally, for all $t < T$ and $\mathbf{x} \in \mathbf{X}$, $f(\mathbf{x})$ is a sample from $\mathcal{GP}_{f,t}$, and $\mathcal{C}_k(\mathbf{x})$ is a sample from $\mathcal{GP}_{\mathcal{C}_k,t}$, for all $k \in \mathbf{K}$.*

**Assumption 2** *A global optimum exists within the feasible region. The distance between this global optimum and the boundaries of the feasible regions is uniformly bounded below by $\epsilon_{\mathcal{C}}$.*

---

[5]Additional discussion in section F

---

**Algorithm 1 COnstrained BO with Adaptive active Learning of unknown constraints (COBALT)**

---

1: **Input**:Search space $\mathbf{X}$, initial observation $\mathbf{D}_0$, horizon $T$, confidence factor $\delta$, estimated $\epsilon_{\mathcal{C}}$;
2: **for** $t = 1\ to\ T$ **do**
3:     Update the posteriors of $\mathcal{GP}_{f,t}$ and $\mathcal{GP}_{\mathcal{C}_k,t}$ according to equation 1 and 2
4:     Identify ROIs $\hat{\mathbf{X}}_t$, and undecided sets $U_{\mathcal{C}_k,t}$
5:     **for** $k \in \mathbf{K}$ **do**
6:         **if** $U_{\mathcal{C}_k,t} \neq \emptyset$ **then**
7:             Candidate for active learning of each constraint:
            $\mathbf{x}_{\mathcal{C}_k,t} \leftarrow \arg\max_{\mathbf{x} \in \tilde{D}_{\hat{\mathbf{X}}_t} \cap U_{\mathcal{C}_k,t}} \alpha_{\mathcal{C}_k,t}(\mathbf{x})$ as in equation 6
8:             $\mathcal{G} \leftarrow \mathcal{G} \cup \mathcal{C}_{k,t}$
9:     Candidate for optimizing the objective: $\mathbf{x}_{f,t} \leftarrow \arg\max_{\mathbf{x} \in \tilde{D}_{\hat{\mathbf{X}}_t}} \alpha_{f,t}(\mathbf{x})$ as in equation 5
10:     $\mathcal{G} \leftarrow \mathcal{G} \cup f$
11:     Maximize the acquisition values from different aspects:
    $g_t \leftarrow \arg\max_{g \in \mathcal{G}} \alpha_{g,t}(\mathbf{x}_{g,t})$
12:     Pick the candidate to evaluate: $\mathbf{x}_t \leftarrow \mathbf{x}_{g_t,t}$
13:     Update the observation set with the candidate and corresponding new observations
    $\mathbf{D}_t \leftarrow \mathbf{D}_{t-1} \cup \{(\mathbf{x}_t, y_{f,t}, \{y_{c_k,t}\}_{k \in \mathbf{K}})\}$

---

*More specifically, for all $k \in \mathbf{K}$, $\exists \epsilon_k > 0$ such that $\mathcal{C}_k(\mathbf{x}^*) > \epsilon_k$, then it holds that $\mathcal{C}_k(\mathbf{x}^*) > \epsilon_{\mathcal{C}} = \min_{k \in \mathbf{K}} \epsilon_k$.*

**Assumption 3** *Given a proper choice of $\beta_t$ that is non-increasing, the confidence interval shrinks monotonically. For all $t_1 < t_2 < T$ and $\mathbf{x} \in \mathbf{X}$, if $\beta_{t_1} \leq \beta_{t_2}$, then $UCB_{t_1}(\mathbf{x}) \geq UCB_{t_2}(\mathbf{x})$ and $LCB_{t_1}(\mathbf{x}) \leq LCB_{t_2}(\mathbf{x})$.*

This is a mild assumption as long as $\beta_t$ is non-increasing, given recent work by Koepernik and Pfaff (2021) showing that if the kernel is continuous and the sequence of sampling points lies sufficiently dense, the variance of the posterior $\mathcal{GP}$ converges to zero almost surely monotonically if the function is in metric space. If the assumption is violated, the technique of taking the intersection of all historical confidence intervals introduced by Gotovos et al. (2013) could similarly guarantee a monotonically shrinking confidence interval. That is, when $\exists t_1 < t_2 < T, \mathbf{x} \in \mathbf{X}$, if we have $UCB_{t_1}(\mathbf{x}) < UCB_{t_2}(\mathbf{x})$ or $LCB_{t_1}(\mathbf{x}) > LCB_{t_2}(\mathbf{x})$, we let $UCB_{t_2}(\mathbf{x}) = UCB_{t_1}(\mathbf{x})$ or $LCB_{t_2}(\mathbf{x}) = LCB_{t_1}(\mathbf{x})$ to guarantee the monotonocity.

The following lemma justifies the definition of the regions(s) of interest $\hat{\mathbf{X}}_t$ defined in equation 4. **For clarity, we denote $\tilde{D}_{\hat{\mathbf{X}}_t} = \tilde{D} \cap \hat{\mathbf{X}}_t$, and $CI_{f^*,t} = [\max_{\mathbf{x} \in \tilde{D}_{\hat{\mathbf{X}}_t}} LCB_t(\mathbf{x}), \max_{\mathbf{x} \in \tilde{D}_{\hat{\mathbf{X}}_t}} UCB_t(\mathbf{x})]$.**

**Lemma 1** *Under the assumptions above, the regions of interest $\hat{\mathbf{X}}_t$, as defined in equation 4, contain the global optimum with high probability. Formally, for all $\delta \in (0,1)$, $T \geq t \geq 1$, and any finite discretization $\tilde{D}$ of $\mathbf{X}$ that contains the optimum $\mathbf{x}^* = \arg\max_{\mathbf{x} \in \mathbf{X}} f(\mathbf{x})$ where $\mathcal{C}_k(\mathbf{x}^*) > \epsilon_{\mathcal{C}}$ for all $k \in \mathbf{K}$ and $\beta_t = 2\log(2(K+1)|\tilde{D}|\pi_t/\delta)$ with $\sum_{T \geq t \geq 1} \pi_t^{-1} = 1$, we have $\mathbb{P}\left[\mathbf{x}^* \in \tilde{D}_{\hat{\mathbf{X}}_t}\right] \geq 1 - \delta$.*

The lemma shows that with proper choice of prior and $\beta$, the $\hat{\mathbf{X}}_{f,t}$ remains nonempty during optimization.

Subsequently, let's define the maximum information gain about function $f$ after $T$ rounds:

$$\gamma_{f,T} = \max_{A \subset \tilde{D}:|A|=T} \mathbb{I}(y_A; f_A) \quad \text{and} \quad \widehat{\gamma_T} = \sum_{g \in \{f\} \cup \{\mathcal{C}_k\}_{k \in \mathbf{K}}} \gamma_{g,T} \quad (7)$$

In the following, we show that we could bound the simple regret of COBALT after sufficient rounds. Concretely, in Theorem 1 we provide an upper bound on the width of the confidence interval for the global optimum $f^* = f(\mathbf{x}^*)$.

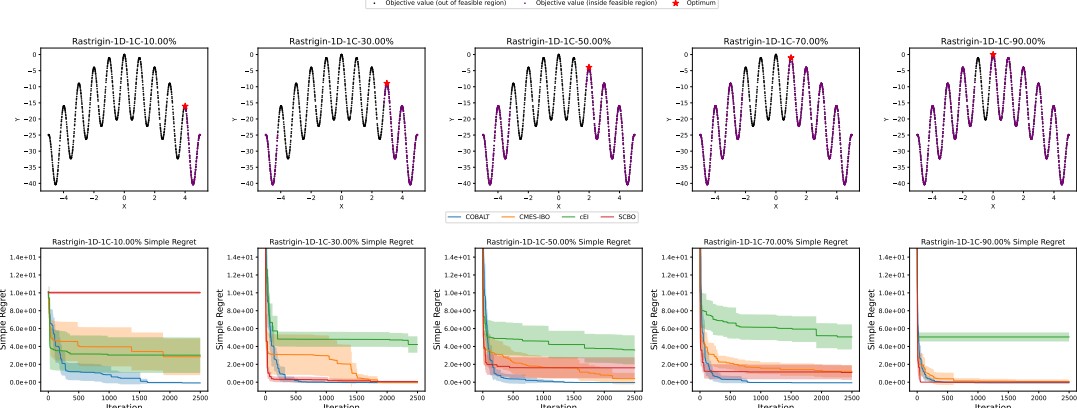

Figure 3: Each column corresponds to a certain threshold choice for the single constraint $c(\mathbf{x}) = |\mathbf{x} + 0.7|^{1/2}$ in the Rastrigin-1D-1C task. The search space contains a certain portion of the feasible region, denoted on each figure and title. The first row shows the distribution of 1000 samples from the noise-free distribution objective function, and the figures are differentiated with different feasible regions. The second row shows corresponding simple regret curves. We test each method with 15 independent trails and impose observation noises sampled from $\mathcal{N}(0, 0.1)$ not shown in the first row.

**Theorem 1** *Under the aforementioned assumptions, with a constant $\beta_t = 2\log\left(\frac{2(K+1)|\tilde{D}|T}{\delta}\right)$ and the acquisition function from $Algorithm$ 1, there exists an $\epsilon \leq \epsilon_{\mathcal{C}}$, such that after at most $T \geq \frac{\beta_T \widehat{\gamma_T} C_1}{\epsilon^2}$ iterations, we have $\mathbb{P}\left[|CI_{f^*,T}| \leq \epsilon, f^* \in CI_{f^*,T}\right] \geq 1 - \delta$ Here, $C_1 = 8/\log(1 + \sigma^{-2})$.*

## 6 EXPERIMENTS

In this section, we empirically study the performance of COBALT against three baselines, including (1) cEI, the extension of EI into CBO from Gelbart et al. (2014), (2) cMES-IBO, a state-of-the-art information-based approach by Takeno et al. (2022), and (3) SCBO, a recent Thompson Sampling (TS) method tailored for scalable CBO from Eriksson and Poloczek (2021). We abstain from comparison against Augmented-Lagrangian methods, following the practice of Takeno et al. (2022), as past studies have illustrated its inferior performance against sampling methods (Eriksson and Poloczek, 2021) or information-based methods (Takeno et al., 2022; Hernández-Lobato et al., 2014). We begin by describing the optimization tasks, and then discuss the performances.

### 6.1 CBO TASKS

We compare COBALT against the aforementioned baselines across six CBO tasks. The first two synthetic CBO tasks are constructed from conventional BO benchmark tasks[6]. Among the other four real-world CBO tasks, the first three are extracted from Tanabe and Ishibuchi (2020), offering a broad selection of multi-objective multi-constraints optimization tasks. The fourth one is a 32-dimensional optimization task extracted from the UCI Machine Learning repository (mis, 2019). Further details about the datasets are available in Appendix D.

- The *Rastrigin function* is a non-convex function used as a performance test problem for optimization algorithms. It was first proposed by Rastrigin (1974) and used as a popular benchmark dataset (Pohlheim). The feasible region takes up approximately 60% of the search space, which we construct by sampling $|\tilde{D}| = 20000$ and reuse for all 15 trials. We also vary the threshold to control the portion of the feasible region to study the robustness of COBALT. Figure 3 shows the distribution of the objective function and feasible regions.

- The *Ackley function* is another commonly used optimization benchmark. We construct two constraints to enforce a feasible area approximately taking up 14% of the search space, which we construct by sampling $|\tilde{D}| = 20000$ and reuse for all 15 trials.

- The *pressure vessel design problem* aims at optimizing the total cost of a cylindrical pressure vessel. The feasible regions take up approximately 78% of the whole search space.

---

[6]Here, we rely on the implementation contained in BoTorch's (Balandat et al., 2020) test function module.

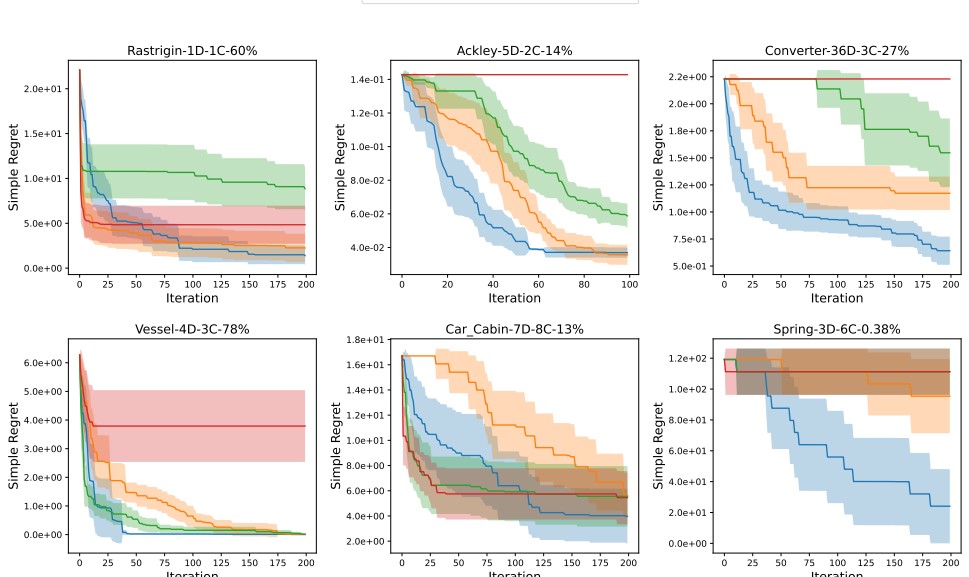

Figure 4: The input dimensionality, the number of constraints, and the approximate portion of the feasible region in the whole search space for each task are denoted on the titles. We run the algorithms on each task for at least 15 independent trials. The curves show the average simple regret after standardization, while the shaded area denotes the 95% confidence interval through the optimization.

- The *coil compression spring design problem* aims to optimize the volume of spring steel wire, which is used to manufacture the spring (Lampinen and Zelinka, 1999) under static loading. The feasible regions take up approximately 0.38% of the whole search space.

- The *car cab design problem* includes seven input variables and eight constraints. The feasible feasible region takes up approximately 13% of the whole search space.

- This *UCI water converter problem* consists of positions and absorbed power outputs of wave energy converters (WECs) from the southern coast of Sydney(mis, 2019). The feasible feasible region takes up approximately 27% of the whole search space.

## 6.2 RESULTS

We study the robustness of the algorithms with varying feasible region sizes on the Rastrigin-1D-1C task. Results are demonstrated in figure 3. Note that the discrete search space consists of the 1000 points shown in the first row of figure 3, and with the observation noises, only COBALT consistently reaches the global optimum within 2000 iterations. The convergence highlights the essential role of the active learning of the constraint in achieving robust optimization when unknown constraints are present.

We further study COBALT on the aforementioned optimization tasks and show the simple regret curves in figure 4. On Rastrigin-1D-1C and Car-Cabin-7D-8C, COBALT lags behind the baselines at the early stage of the optimization, potentially due to the active learning of the constraints outweighing the optimization. The steady improvement of COBALT through the optimization allows a consistently superior performance after sufficient iterations. In contrast, the baselines are trapped at the local optimum on these two tasks. The results show that COBALT is efficient and effective in various settings, including different dimensionalities of input space, different numbers of constraints, and different correlations between constraints.

## 7 CONCLUSION

Bayesian optimization with unknown constraints poses challenges in the adaptive tradeoff between optimizing the unknown objective and learning the constraints. We introduce COBALT, which is backed by rigorous theoretical guarantees, to efficiently address constrained Bayesian optimization. Our key insights include: (1) the ROIs determined through adaptive level-set estimation can congregate and contribute to the overall Bayesian optimization task; (2) acquisition functions based on independent GPs can be unified in a principled way. Through extensive experiments, we validate the efficacy and robustness of our proposed method across various optimization tasks.

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

# A PROOFS

## A.1 PROOF OF LEMMA 1

**Lemma 1** *Under the assumptions above, the regions of interest $\hat{\mathbf{X}}_t$, as defined in equation 4, contain the global optimum with high probability. Formally, for all $\delta \in (0,1)$, $T \geq t \geq 1$, and any finite discretization $\tilde{D}$ of $\mathbf{X}$ that contains the optimum $\mathbf{x}^* = \arg\max_{\mathbf{x} \in \mathbf{X}} f(\mathbf{x})$ where $\mathcal{C}_k(\mathbf{x}^*) > \epsilon_{\mathcal{C}}$ for all $k \in \mathbf{K}$ and $\beta_t = 2\log(2(K+1)|\tilde{D}|\pi_t/\delta)$ with $\sum_{t\geq 1}^T \pi_t^{-1} = 1$, we have $\mathbb{P}\left[\mathbf{x}^* \in \tilde{D}_{\hat{\mathbf{X}}_t}\right] \geq 1 - \delta$.*

*Proof:* Similar to lemma 5.1 of Srinivas et al. (2009), with probability at least $1 - 1/2\delta$, $\forall \mathbf{x} \in \tilde{D}, \forall T \geq t \geq 1, \forall g \in \{f\} \cup \{\mathcal{C}_k\}_{k \in \mathbf{K}}$,

$$|g(\mathbf{x}) - \mu_{g,t-1}(\mathbf{x})| \leq \beta_t^{1/2}\sigma_{g,t-1}(\mathbf{x})$$

Note that we also take the union bound on $g \in \{f\} \cup \{\mathcal{C}_k\}_{k \in \mathbf{K}}$.

**First, by definition $S_{\mathcal{C},t} \triangleq \bigcap_k^{\mathbf{K}} S_{\mathcal{C}_k,t}$, we have $\forall t \leq T, \mathbf{x} \in \tilde{D} \cap S_{\mathcal{C},t}, \forall k \in \mathbf{K}$**

$$\mathbb{P}\left[\mathcal{C}_k(\mathbf{x}) \geq \text{LCB}_{\mathcal{C}_k,t}(\mathbf{x}) = \mu_{\mathcal{C}_k,t-1}(\mathbf{x}) - \beta_t^{1/2}\sigma_{\mathcal{C}_k,t-1}(\mathbf{x}) > 0\right] \geq 1 - 1/2\delta$$

meaning with probability at $1 - \delta$, $\mathbf{x}$ lies in the feasible region. At the same time, we have, $\forall t \leq T$

$$\mathbb{P}\left[\text{UCB}_{f,t}(\mathbf{x}^*) \geq f(\mathbf{x}^*) \geq f(\mathbf{x}) \geq \text{LCB}_{f,t}(\mathbf{x}) \mid \mathcal{C}_k(\mathbf{x}) > 0, \forall k \in \mathbf{K}\right] \geq 1 - 1/2\delta$$

Given the mutual independency between the objective $f$ and the constraints $\mathcal{C}_k$, and by the definition of the threshold $\text{LCB}_{f,t,max}$, we have $\forall t \leq T$, **when $\exists \mathbf{x} \in \tilde{D} \cap S_{\mathcal{C},t}$,**

$$\mathbb{P}\left[\text{UCB}_{f,t}(\mathbf{x}^*) > \text{LCB}_{f,t,max}\right] \geq (1 - 1/2\delta)^2 \geq 1 - \delta$$

Note when $\tilde{D} \cap S_{\mathcal{C},t} = \emptyset$, $\text{LCB}_{f,t,max} = -\infty$, we have $\mathbb{P}\left[\text{UCB}_{f,t}(\mathbf{x}^*) > \text{LCB}_{f,t,max}\right] = 1$.

**In summary, we've shown that with probability at least $1 - \delta$, $\mathbf{x}^* \in \tilde{D} \cap \hat{\mathbf{X}}_{f,t}$.**

Next, by the definition of $\mathbf{x}^* = \arg\max_{\mathbf{x} \in \mathbf{X}} f(\mathbf{x})$ $s.t.$ $\mathcal{C}_k(\mathbf{x}^*) > \epsilon_{\mathcal{C}}$ we have $\forall t \leq T, \forall k \in \mathbf{K}$

$$\mathbb{P}\left[\text{UCB}_{\mathcal{C}_k,t}(\mathbf{x}^*) = \mu_{\mathcal{C}_k,t-1}(\mathbf{x}^*) + \beta_t^{1/2}\sigma_{\mathcal{C}_k,t-1}(\mathbf{x}^*) \geq \mathcal{C}_k(\mathbf{x}^*) > 0\right] \geq 1 - 1/2\delta$$

**meaning with probability at least $1 - 1/2\delta$, $\mathbf{x}^* \in \tilde{D} \cap \hat{\mathbf{X}}_{\mathcal{C}_k,t}$. And in general, we have $\forall t \leq T, \forall k \in \mathbf{K}$**

$$\mathbb{P}\left[\mathbf{x}^* \in \tilde{D} \cap \hat{\mathbf{X}}_t\right] \geq 1 - \delta$$

**Note that different from lemma 5.1 of Srinivas et al. (2009), we do not require the lemma hold for $\forall t \geq 1$, instead we require it to hold for $\forall T \geq t \geq 1$. This alleviates the need of the convergence of the series $\sum_{t\geq 1} \pi_t^{-1} = 1$ to $\sum_{t\geq 1}^T \pi_t^{-1} = 1$ when taking the union bound. Specifically, we could set $\pi_t = T$, which essentially makes $\beta_t = 2\log(\frac{2(K+1)|\tilde{D}|T}{\delta})$ a constant. Hence, we use the $\beta$ in the following instead of $\beta_t$ as traditionally used to highlight this difference.** $\qquad \square$

## A.2 PROOF OF THEOREM 1

The following lemmas show that the maximum of the acquisition functions equation 5 and 6 are both bounded after sufficient evaluations.

**Lemma A.1** *Under the conditions assumed in Theorem 1 except for Assumption 2, let $\alpha_t = \max_{g \in \mathcal{G}} \alpha_{g,t}(\mathbf{x}_{g,t})$ as in Algorithm 1, with $\beta = 2\log(\frac{2(K+1)|\tilde{D}|T}{\delta})$ that is a constant, after at most $T \geq \frac{\beta \hat{\gamma}_T C_1}{\epsilon^2}$ iterations, $\alpha_T \leq \epsilon$ Here $C_1 = 8/\log(1 + \sigma^{-2})$.*

**The inequation $T \geq \frac{\beta \widehat{\gamma_T} C_1}{\epsilon^2}$ has $T$ on both side, which follows the convention in Gotovos et al. (2013).**

*Proof:* **We first unify the notation in the acquisition functions.**
$\forall T \geq t \geq 1, \forall g \in \{\mathcal{C}_k\}_{k \in \mathbf{K}}$, when $\tilde{D}_{\hat{\mathbf{X}}_t} \cap U_{g,t} \neq \emptyset$,

$$\max_{\mathbf{x} \in \tilde{D}_{\hat{\mathbf{X}}_t} \cap U_{g,t}} \text{UCB}_{g,t}(\mathbf{x}) - \text{LCB}_{g,t}(\mathbf{x}) = 2\beta^{1/2}\sigma_{g,t-1}(\mathbf{x}_{g,t}) \leq \alpha_t \tag{8}$$

$\forall T \geq t \geq 1, \forall g \in \{\mathcal{C}_k\}_{k \in \mathbf{K}}$, when $\tilde{D}_{\hat{\mathbf{X}}_t} \cap U_{\mathcal{C}_k,t} = \emptyset$, let

$$\max_{\mathbf{x} \in \tilde{D}_{\hat{\mathbf{X}}_t} \cap U_{g,t}} \text{UCB}_{g,t}(\mathbf{x}) - \text{LCB}_{g,t}(\mathbf{x}) = 2\beta^{1/2}\sigma_{g,t-1}(\mathbf{x}_{g,t}) = 0 \leq \alpha_t \tag{9}$$

$\forall T \geq t \geq 1, g = f$

$$\max_{\mathbf{x} \in \tilde{D}_{\hat{\mathbf{X}}_t}} \text{UCB}_{f,t}(\mathbf{x}) - \text{LCB}_{f,t,max} \leq \text{UCB}_{f,t}(\mathbf{x}_{f,t}) - \text{LCB}_{f,t}(\mathbf{x}_{f,t}) \tag{10}$$

$$= 2\beta^{1/2}\sigma_{f,t-1}(\mathbf{x}_{f,t}) \tag{11}$$

$$\leq \alpha_t \tag{12}$$

**By lemma 5.1, 5.2 and 5.4 of Srinivas et al. (2009), with $\beta = 2\log(\frac{2(K+1)|\tilde{D}|T}{\delta})$, $\forall g \in \{f\} \cup \{\mathcal{C}_k\}_{k \in \mathbf{K}}$ and $\forall x_t \in \tilde{D}_{\hat{\mathbf{X}}_t} \subseteq \tilde{D}$, we have $\sum_{t=1}^{T}(2\beta^{1/2}\sigma_{g,t-1},(\mathbf{x}_t))^2 \leq C_1\beta\gamma_{g,T}$. By definition of $\alpha_t$ , we have the following**

$$\sum_{t=1}^{T}\alpha_t^2 \leq \sum_{t=1}^{T}\sum_{g \in \{\mathcal{C}_k\}_{k \in \mathbf{K}}}(\alpha_{g,t}(\mathbf{x}_t))^2$$

$$\leq \sum_{t=1}^{T}\sum_{g \in \{\mathcal{C}_k\}_{k \in \mathbf{K}}}(2\beta^{1/2}\sigma_{g,t-1}(\mathbf{x}_t))^2$$

$$\leq \sum_{g \in \{\mathcal{C}_k\}_{k \in \mathbf{K}}}C_1\beta\gamma_{g,T}$$

$$= C_1\beta\widehat{\gamma_T}$$

The last line holds due to the definition in equation 7. By Cauchy-Schwarz, we have

$$\frac{1}{T}(\sum_{t=1}^{T}\alpha_t)^2 \leq C_1\beta\widehat{\gamma_T}$$

**By the monotonocity assumed in** *Assumption 3,* **$\forall g \in \{\mathcal{C}_k\}_{k \in \mathbf{K}}$, $\forall 1 \leq t_1 < t_2 \leq T$, $\forall g \in \{\mathcal{C}_k\}_{k \in \mathbf{K}}$, we have $U_{g,t_2} \subseteq U_{g,t_1}$ and $\hat{\mathbf{X}}_{t_2} \subseteq \hat{\mathbf{X}}_{t_1}$, and most importantly, $\alpha_{t_2} \leq \alpha_{t_1}$. Therefore**

$$\alpha_T \leq \frac{1}{T}\sum_{t=1}^{T}\alpha_t \leq \sqrt{\frac{C_1\beta\widehat{\gamma_T}}{T}}$$

**As a result, after at most $T \geq \frac{\beta\widehat{\gamma_T}C_1}{\epsilon^2}$ iterations, we have $\alpha_T \leq \epsilon$.**

$\square$

With Lemma A.1, we could first prove that after adequately $T$ rounds of evaluations such that $\epsilon \leq \min_{k \in \mathbf{K}} \epsilon_k$ is sufficiently small, with certain probability, $\mathbf{x}^* \in S_{\mathcal{C},T}$. Then $\text{LCB}_{f,t,max} \neq -\infty$, and therefore the width of $[\max_{\mathbf{x} \in \tilde{D}_{\hat{\mathbf{X}}_t}} \text{LCB}_{f,T}(\mathbf{x}), \max_{\mathbf{x} \in \tilde{D}_{\hat{\mathbf{X}}_t}} \text{UCB}_{f,T}(\mathbf{x})]$, which is a the high confidence interval of $f^*$, is bounded by $\epsilon$.

*Proof:* We first prove that after at most $T \geq \frac{\beta\widehat{\gamma_T}C_1}{\epsilon^2}$ iterations, $\mathbb{P}\left[\mathbf{x}^* \in \tilde{D}_{\hat{\mathbf{X}}_t} \cap S_{\mathcal{C},T}\right] \geq 1 - 1/2\delta$. Given equation 8 and 9 and Lemma A.1, we have $\forall g \in \{\mathcal{C}_k\}_{k \in \mathbf{K}}$,

$$\max_{\mathbf{x} \in \tilde{D}_{\hat{X}_T} \cap U_{g,T}} \text{UCB}_{g,T}(\mathbf{x}) - \text{LCB}_{g,T}(\mathbf{x}) \leq \epsilon \leq \min_{k \in \mathbf{K}} \epsilon_k$$

According to the definition of $U_{g,T}$, $\forall \mathbf{x} \in \tilde{D}_{\hat{X}_T} \cap U_{g,T}$, $\forall g \in \{\mathcal{C}_k\}_{k \in \mathbf{K}}$

$$\text{UCB}_{g,T}(\mathbf{x}) \leq \min_{k \in \mathbf{K}} \epsilon_k + \text{LCB}_{g,T}(\mathbf{x}) \leq \min_{k \in \mathbf{K}} \epsilon_k$$

According to Assumption 2, and Lemma 1, we have $\forall k \in \mathbf{K}$

$$\mathbb{P}\left[\text{UCB}_{\mathcal{C}_k,T}(\mathbf{x}^*) \geq \mathcal{C}_k(\mathbf{x}^*) > \epsilon_k \geq \max_{\mathbf{x} \in \tilde{D}_{\hat{\mathbf{X}}_t} \cap U_{\mathcal{C}_k,t}} \text{UCB}_{\mathcal{C}_k,T}(\mathbf{x})\right] \geq 1 - 1/2\delta$$

Hence when $t = T$

$$\mathbb{P}\left[\mathbf{x}^* \in \tilde{D}_{\hat{X}_T} \cap S_{\mathcal{C},T} = \tilde{D}_{\hat{\mathbf{X}}_t} \cap \hat{\mathbf{X}}_{\mathcal{C},T} \setminus \cup_{k \in \mathbf{K}} U_{\mathcal{C}_k,T}\right] \geq 1 - 1/2\delta \tag{13}$$

As a result
$$\mathbb{P}\left[\text{LCB}_{f,T,max} \neq -\infty\right] \geq 1 - 1/2\delta$$
Next, we prove the upper bound for the width of the high-confidence interval of $f^*$. Given that $\text{LCB}_{f,T,max} \neq -\infty$, we have

$$\max_{\mathbf{x} \in \tilde{D}_{\hat{X}_T}} \text{UCB}_{f,T}(\mathbf{x}) - \max_{\mathbf{x} \in \tilde{D}_{\hat{X}_T}} \text{LCB}_{f,T}(\mathbf{x}) \leq \max_{\mathbf{x} \in \tilde{D}_{\hat{X}_T}} \text{UCB}_{f,T}(\mathbf{x}) - \text{LCB}_{f,T,max}$$
$$\leq \alpha_T$$
$$\leq \epsilon$$

Combining it with the fact that

$$\mathbb{P}\left[\max_{\mathbf{x} \in \tilde{D}_{\hat{X}_T}} \text{LCB}_{f,T}(\mathbf{x}) \leq \max_{\mathbf{x} \in \tilde{D}_{\hat{X}_T}} f(\mathbf{x}) = f^* \leq \text{UCB}_{f,T}(\mathbf{x}^*) \leq \max_{\mathbf{x} \in \tilde{D}_{\hat{X}_T}} \text{UCB}_{f,T}(\mathbf{x})\right] \geq 1 - 1/2\delta$$

we attain the final result that after $T \geq \frac{\beta \widehat{\gamma_T} C_1}{\epsilon^2}$ iterations,

$$\mathbb{P}\left[|CI_{f^*,T}| \leq \epsilon, f^* \in CI_{f^*,T}\right] \geq 1 - \delta$$

$\square$

**Similar to the proof of Theorem 1, with Lemma A.1, we can show that the algorithm can identify infeasibility when all points in the search space violate at least one of the constraints at least $\epsilon'_{\mathcal{C}}$. Concretely, $\forall \mathbf{x} \in \mathbf{X}$, if it holds that $\exists k \in \mathbf{K}, \mathcal{C}_k(x) < -\epsilon'_{\mathcal{C}}$, with high probability the identified $\tilde{D}_{\hat{\mathbf{X}}_T} = \emptyset$.**

**Corollary 1** *If the assumptions except for Assumption 2 hold, $\forall \mathbf{x} \in \mathbf{X}$, if it holds that $\exists k \in \mathbf{K}$, $\mathcal{C}_k(x) < -\epsilon'_{\mathcal{C}}$. Then with a constant $\beta = 2\log(\frac{2(K+1)|\tilde{D}|T}{\delta})$ and the acquisition function from Algorithm 1, after at most $T \geq \frac{\beta \widehat{\gamma_T} C_1}{\epsilon'^2_{\mathcal{C}}}$ iterations, we have $\mathbb{P}\left[\tilde{D}_{\hat{\mathbf{X}}_T} = \emptyset\right] \geq 1 - \delta$. Here, $C_1 = 8/\log(1 + \sigma^{-2})$.*

*Proof:* We assume $\tilde{D}_{\hat{X}_T} \neq \emptyset$ and prove by contradiction. Given equation 8 and 9 and Lemma A.1, we have $\forall g \in \{\mathcal{C}_k\}_{k \in \mathbf{K}}$,

$$\max_{\mathbf{x} \in \tilde{D}_{\hat{X}_T} \cap U_{g,T}} \text{UCB}_{g,T}(\mathbf{x}) - \text{LCB}_{g,T}(\mathbf{x}) \leq \epsilon'_{\mathcal{C}}$$

According to the definition of $U_{g,T}$, $\forall \mathbf{x} \in \tilde{D}_{\hat{X}_T} \cap U_{g,T}$, $\forall g \in \{\mathcal{C}_k\}_{k \in \mathbf{K}}$

$$\text{UCB}_{g,T}(\mathbf{x}) \leq \epsilon'_{\mathcal{C}} + \text{LCB}_{g,T}(\mathbf{x})$$

Then we have $\forall \mathbf{x} \in \tilde{D}_{\hat{X}_T} \cap U_{g,T}$, $\exists k \in \mathbf{K}$

$$\mathbb{P}\left[\mathcal{C}_k(\mathbf{x}) \leq \text{UCB}_{\mathcal{C}_k,T}(\mathbf{x}) \leq \epsilon'_{\mathcal{C}} + \text{LCB}_{g,T}(\mathbf{x}) \leq \epsilon'_{\mathcal{C}} + \mathcal{C}_k(\mathbf{x}) < 0\right] \geq 1 - 1/2\delta$$

This contradiction means $\forall g \in \{\mathcal{C}_k\}_{k \in \mathbf{K}}$, $\tilde{D}_{\hat{X}_T} \cap U_{g,T} = \emptyset$ with probability as least $1 - 1/2\delta$.

According to the definition of $S_{g,T}$, $\forall \mathbf{x} \in \tilde{D}_{\hat{X}_T} \cap S_{g,T}, \forall g \in \{\mathcal{C}_k\}_{k \in \mathbf{K}}$

$$\text{LCB}_{g,T}(\mathbf{x}) \geq \epsilon'_{\mathcal{C}}$$

Then we have $\forall \mathbf{x} \in \tilde{D}_{\hat{X}_T} \cap S_{g,T}, \exists g \in \{\mathcal{C}_k\}_{k \in \mathbf{K}}$

$$\mathbb{P}\left[-\epsilon'_{\mathcal{C}} \geq \mathcal{C}_k(\mathbf{x}) \geq \text{LCB}_{g,T}(\mathbf{x}) \geq \epsilon'_{\mathcal{C}}\right] \geq 1 - 1/2\delta$$

This contradiction means $\forall g \in \{\mathcal{C}_k\}_{k \in \mathbf{K}}$, $\tilde{D}_{\hat{\mathbf{X}}_t} \cap S_{g,T} = \emptyset$ with probability as least $1 - 1/2\delta$.

Combining the above contradictions, we have at least when $t = T$,

$$\mathbb{P}\left[\tilde{D}_{\hat{\mathbf{X}}_T} = \emptyset\right] \geq 1 - \delta$$

$\square$

**Another direct result of Theorem 1 is that, if at $T + 1$, the algorithm picks $\arg\max_{\mathbf{x} \in \tilde{D}_{\hat{X}_T} \cap S_{\mathcal{C},T}} \text{LCB}_{f,T}(\mathbf{x})$, the evaluation translates Theorem 1 into high probability bound of simple regret .**

**Corollary 2** *Under the assumptions of Theorem 1, with a constant $\beta = 2\log\left(\frac{2(K+1)|\tilde{D}|(T+1)}{\delta}\right)$ and the acquisition function from Algorithm 1, there exists an $\epsilon \leq \min_{k \in \mathbf{K}} \epsilon_k$, such that when $T \geq \frac{\beta \widehat{\gamma_T} C_1}{\epsilon^2}$, and the algorithm pick $\mathbf{x}_{T+1} = \arg\max_{\mathbf{x} \in \tilde{D}_{\hat{\mathbf{X}}_T} \cap S_{\mathcal{C},T}} LCB_{f,T}(\mathbf{x})$, we have $\mathbb{P}\left[\mathbf{R}_{T+1} \leq \epsilon\right] \geq 1 - \delta$. Here, $C_1 = 8/\log(1 + \sigma^{-2})$.*

*Proof:* We omit the shared part with the proof of Theorem 1. **With $\beta = 2\log\left(\frac{2(K+1)|\tilde{D}|(T+1)}{\delta}\right) > 2\log\left(\frac{2(K+1)|\tilde{D}|T}{\delta}\right)$, the bound hold for the original analysis.** After attaining 13 and

$$\mathbb{P}\left[\text{LCB}_{f,T,max} \neq -\infty\right] \geq 1 - 1/2\delta$$

We prove the upper bound of the simple regret. Given that $\text{LCB}_{f,T,max} \neq -\infty$, we have

$$\max_{\mathbf{x} \in \tilde{D}_{\hat{X}_T} \cap S_{\mathcal{C},t}} \text{UCB}_{f,T}(\mathbf{x}) - \text{LCB}_{f,T}(\mathbf{x}_{T+1}) \leq \max_{\mathbf{x} \in \tilde{D}_{\hat{X}_T}} \text{UCB}_{f,T}(\mathbf{x}) - \text{LCB}_{f,T,max}$$
$$\leq \alpha_T$$
$$\leq \epsilon$$

Note 13 shows that with probability at least $1 - \delta/2$, $\mathbf{x}^* \in \tilde{D}_{\hat{\mathbf{X}}_T} \cap S_{\mathcal{C},T}$. And by the definition of $\tilde{D}_{\hat{\mathbf{X}}_T} \cap S_{\mathcal{C},T}$, we attain the final simple regret bound that as long as $T \geq \frac{\beta \widehat{\gamma_T} C_1}{\epsilon^2}$, at $T + 1$, we have

$$\mathbb{P}\left[\mathbf{R}_{T+1} \leq f^* - f(\mathbf{x}_{T+1}) \leq \max_{\mathbf{x} \in \tilde{D}_{\hat{X}_T} \cap S_{\mathcal{C},T}} \text{UCB}_{f,T}(\mathbf{x}) - \text{LCB}_{f,T,max} \leq \epsilon\right] \geq 1 - \delta$$

$\square$

# B    DECOUPLED SETTING

In the main paper, we assume both objective $f$ and the constraints $\{\mathcal{C}_k\}_{k \in \mathbf{K}}$ are revealed upon querying an input point. The setting is regarded as a coupling of the objective and constraints to differentiate from the decoupled setting, where the objective and constraints may be evaluated independently. In the decoupled setting, acquisition functions need to explicitly tradeoff the evaluation of the different aspects and, in addition to helping to pick the candidate $\mathbf{x}_t \in \mathbf{X}$, suggest $g_t \in \{f\} \cup \{\mathcal{C}_k\}_{k \in \mathbf{K}}$ for evaluation each time. This typically requires different acquisition from coupled setting (Gelbart et al., 2014). However, we will show that our acquisition function and COBALT require minimum adaptation to the decoupled setting while bearing a similar performance guarantee.

**Algorithm 2 COnstrained BO with Adaptive active Learning of *decoupled* unknown constraints (COBALT-Decoupled)**

1: **Input:** Search space $\mathbf{X}$, initial observation $\mathbf{D}_0$, horizon $T$, confidence factor $\delta$, estimated $\epsilon_{\mathcal{C}}$;
2: **for** $t = 1$ *to* $T$ **do**
3:    Update the posteriors of $\mathcal{GP}_{f,t}$ and $\mathcal{GP}_{\mathcal{C}_k,t}$ according to equation 1 and 2
4:    Identify ROIs $\hat{\mathbf{X}}_t$, and undecided sets $U_{\mathcal{C}_k,t}$
5:    **for** $k \in \mathbf{K}$ **do**
6:      **if** $U_{\mathcal{C}_k,t} \neq \emptyset$ **then**
7:        Candidate for active Learning of each constraint:
         $\mathbf{x}_{\mathcal{C}_k,t} \leftarrow \arg\max_{\mathbf{x} \in \tilde{D}_{\hat{\mathbf{X}}_t} \cap U_{\mathcal{C}_k,t}} \alpha_{\mathcal{C}_k,t}(\mathbf{x})$ as in equation 6
8:        $\mathcal{G} \leftarrow \mathcal{G} \cup \mathcal{C}_{k,t}$
9:    Candidate for optimizing the objective:
      $\mathbf{x}_{f,t} \leftarrow \arg\max_{\mathbf{x} \in \tilde{D}_{\hat{\mathbf{X}}_t}} \alpha_{f,t}(\mathbf{x})$ as in equation 5
10:    $\mathcal{G} \leftarrow \mathcal{G} \cup f$
11:   Maximize the acquisition values from different aspects:
      $g_t \leftarrow \arg\max_{g \in \mathcal{G}} \alpha_{g,t}(\mathbf{x}_{g_t,t})$
12:   Pick the candidate to evaluate: $\mathbf{x}_t \leftarrow \mathbf{x}_{g_t,t}$
13:   *Update the observation set with the candidate and corresponding new observations on $g_t$*
      $\mathbf{D}_t \leftarrow \mathbf{D}_{t-1} \cup \{(\mathbf{x}_t, y_{g_t,t})\}$

## B.1 Algorithm for Decoupled Setting

When taking the $g_t \leftarrow \arg\max_{g \in \mathcal{G}} \alpha_{g,t}(\mathbf{x}_{g,t})$ in Algorithm 1, we explicitly choose the aspect that matters most at a certain iteration. Naturally, we could adapt COBALT to the decoupled setting by querying $\mathbf{x}_{g,t}$ on this unknown function $g_t \in \mathcal{G} \subseteq \{f\} \cup \{\mathcal{C}_k\}_{k \in \mathbf{K}}$ at iteration $t$. The modified algorithm is shown below.

## B.2 Theoretical guarantee and proof

We first denote the maximum mutual information gain after $T$ rounds of evaluations as

$$\widetilde{\gamma}_T = \sum_{g \in \{f\} \cup \{\mathcal{C}_k\}_{k \in \mathbf{K}}} \gamma_{g,T_g} \tag{14}$$

Where $T_g$ denotes the number of evaluations for $g \in \{f\} \cup \{\mathcal{C}_k\}_{k \in \mathbf{K}}$ before $T$. Therefore we have

$$T = \sum_{g \in \{f\} \cup \{\mathcal{C}_k\}_{k \in \mathbf{K}}} T_g$$

Then, we have the following guarantee for the performance of COBALT-Decoupled.

**Theorem 2** *The width of the resulting confidence interval of the global optimum $f^* = f(\mathbf{x}^*)$ has an upper bound. That is, under the same assumptions in Theorem 1, with $\beta = 2\log(2(K + 1)|\tilde{D}_{\hat{\mathbf{X}}_t}|\pi_t/\delta)$ that is constant, and acquisition function in $Algorithm\ 2$, $\exists \epsilon \leq \epsilon_{\mathcal{C}}$, after at most $T \geq \frac{\beta\widetilde{\gamma}_T C_1}{\epsilon^2}$ iterations, we have $\mathbb{P}\left[|CI_{f^*,T}| \leq \epsilon, f^* \in CI_{f^*,T}\right] \geq 1 - \delta$ Here $C_1 = 8/\log(1 + \sigma^{-2})$.*

**Lemma B.1** *Under the conditions assumed in Theorem 2 except for Assumption 2, let $\alpha_t = \max_{g \in \mathcal{G}} \alpha_{g,t}(\mathbf{x}_{g,t})$ as in Algorithm 2, with $\beta = 2\log(\frac{2(K+1)|\tilde{D}|T}{\delta})$ that is a constant, after at most $T \geq \frac{\beta\widetilde{\gamma}_T C_1}{\epsilon^2}$ iterations, $\alpha_T \leq \epsilon$ Here $C_1 = 8/\log(1 + \sigma^{-2})$.*

**Here is the critical difference to the proof of Theorem 1.**

*Proof:* **We first unify the notation in the acquisition functions.**
$\forall T \geq t \geq 1, \forall g \in \{\mathcal{C}_k\}_{k \in \mathbf{K}}$, when $\tilde{D}_{\hat{\mathbf{X}}_t} \cap U_{g,t} \neq \emptyset$,

$$\max_{\mathbf{x} \in \tilde{D}_{\hat{\mathbf{X}}_t} \cap U_{g,t}} \mathrm{UCB}_{g,t}(\mathbf{x}) - \mathrm{LCB}_{g,t}(\mathbf{x}) = 2\beta^{1/2}\sigma_{g,t-1}(\mathbf{x}_{g,t}) \leq \alpha_t \tag{15}$$

$\forall T \geq t \geq 1, \forall g \in \{\mathcal{C}_k\}_{k \in \mathbf{K}}$, when $\tilde{D}_{\hat{\mathbf{X}}_t} \cap U_{\mathcal{C}_k, t} = \emptyset$, let

$$\max_{\mathbf{x} \in \tilde{D}_{\hat{\mathbf{X}}_t} \cap U_{g,t}} \text{UCB}_{g,t}(\mathbf{x}) - \text{LCB}_{g,t}(\mathbf{x}) = 2\beta^{1/2}\sigma_{g,t-1}(\mathbf{x}_{g,t}) = 0 \leq \alpha_t \tag{16}$$

$\forall T \geq t \geq 1, g = f$

$$\max_{\mathbf{x} \in \tilde{D}_{\hat{\mathbf{X}}_t}} \text{UCB}_{f,t}(\mathbf{x}) - \text{LCB}_{f,t,max} \leq \text{UCB}_{f,t}(\mathbf{x}_{f,t}) - \text{LCB}_{f,t}(\mathbf{x}_{f,t}) \tag{17}$$

$$= 2\beta^{1/2}\sigma_{f,t-1}(\mathbf{x}_{f,t}) \tag{18}$$

$$\leq \alpha_t \tag{19}$$

**By lemma 5.1, 5.2 and 5.4 of Srinivas et al. (2009), with** $\beta = 2\log(\frac{2(K+1)|\tilde{D}|T}{\delta})$, $\forall g \in \{f\} \cup$ $\{\mathcal{C}_k\}_{k \in \mathbf{K}}$ **and** $\forall x_t \in \tilde{D}_{\hat{\mathbf{X}}_t} \subseteq \tilde{D}$**, we have** $\sum_{t=1}^{T}(2\beta^{1/2}\sigma_{g,t-1}, (\mathbf{x}_t))^2 \mathbf{1}(g_t = g) \leq C_1\beta\gamma_{g,T_g}$**. By definition of** $\alpha_t$ **, we have the following**

$$\sum_{t=1}^{T}\alpha_t^2 = \sum_{t=1}^{T}\alpha_{g_t,t}^2(\mathbf{x}_{g_t,t})$$
$$\leq \sum_{t=1}^{T}(2\beta^{1/2}\sigma_{g_t,t-1}(\mathbf{x}_{g_t,t}))^2$$
$$\leq \sum_{g \in \mathcal{G}} C_1\beta\gamma_{g,T_g}$$
$$= C_1\beta\widetilde{\gamma}_T$$

By Cauchy-Schwarz, we have

$$\frac{1}{T}(\sum_{t=1}^{T}\alpha_t)^2 \leq C_1\beta\widetilde{\gamma}_T$$

By the monotonocity assumed in *Assumption* 3, $\forall g \in \{\mathcal{C}_k\}_{k \in \mathbf{K}}$, $\forall 1 \leq t_1 < t_2 \leq T$, $\forall g \in \{\mathcal{C}_k\}_{k \in \mathbf{K}}$, we have $U_{g,t_2} \subseteq U_{g,t_1}$ and $\hat{\mathbf{X}}_{t_2} \subseteq \hat{\mathbf{X}}_{t_1}$, and most importantly, $\alpha_{t_2} \leq \alpha_{t_1}$. Therefore

$$\alpha_T \leq \frac{1}{T}\sum_{t=1}^{T}\alpha_t \leq \sqrt{\frac{C_1\beta\widetilde{\gamma}_T}{T}}$$

As a result, after at most $T \geq \frac{\beta\widetilde{\gamma}_T C_1}{\epsilon^2}$ iterations, we have $\alpha_T \leq \epsilon$. $\qquad\square$

**The rest of the proof for Theorem 2 is essentially the same as proof for Theorem 1 except for substituting Lemma A.1 with Lemma B.1.**

## C  REWARD FUNCTION

### C.1  REWARD CHOICE 1: PRODUCT OF REWARD AND FEASIBILITY

The definition of reward plays an important role in online machine learning performance analysis. In the CBO setting, one possible definition of constrained reward derived from the constraint nature is $r(\mathbf{x}) = f(\mathbf{x})\prod_k \mathbb{1}_{\mathcal{C}_k(\mathbf{x})>h_k}$ when assuming the $f(\mathbf{x}) > 0$. Considering both the aleatoric and epistemic uncertainty on the constraints, we could transform the problem into finding the maximizer

$$\arg\max_{\mathbf{x} \in \mathbf{X}} r(\mathbf{x}) = \arg\max_{\mathbf{x} \in \mathbf{X}} f(\mathbf{x})\prod_k \mathbb{P}\left[Y_{\mathcal{C}_k}(\mathbf{x}) > h_k\right]$$

Here $Y_{\mathcal{C}_k}(\mathbf{x})$ denotes the observation of the constraint $\mathcal{C}_k$ at $\mathbf{x}$.

The problem with this product reward, on the one hand, is that it is likely to incur a Pareto front if we regard the problem as a multi-objective optimization where the objectives are composed of $f(\mathbf{x})$ and $\mathbb{P}\left[Y_{\mathcal{C}_k}(\mathbf{x}) > h_k\right]$. The multi-objective nature and resulting Pareto front indicate that the optimization could be more challenging to converge than the single-objective unconstrained BO problem, though the unique global optimum is not always expected there either. More critically, when the feasibility of reaching a certain threshold, we prefer to focus on optimizing the objective value rather than the product for the following reasons.

Firstly, the marginal gain on improving feasibility by increasing the value of the constraint function drops after the feasibility reaches 0.5 if assuming it follows a Gaussian. Especially in the tail region, improving the feasibility and then the product of feasibility and objective value by optimizing the constraint function is prohibitively difficult.

Secondly, in most real-world scenarios except for certain applications that focus on feasibility (where the feasibility should be treated as another objective and make it in nature a multi-objective optimization), the actual marginal gain, in general, increases the feasibility decay faster than the increase of objective value. (e.g., when choosing between doubling the feasibility from 0.25 to 0.5 or doubling the objective drop from 25 to 50, we probably favor the former as 0.25, meaning it is unlikely to happen. However, when choosing between increasing feasibility from .8 to .9 or increasing the objective drop from 80 to 90, there would be no such clear preference.) Then, the user would possibly favor the gain on the objective function after the feasibility reaches a certain level. Therefore, we propose the following reward for constrained optimization tasks according to this insight.

### C.2 REWARD CHOICE 2: OBJECTIVE FUNCTION AFTER THE FEASIBILITY REACHING CERTAIN THRESHOLD

Instead of defining the reward as the product of the objective value and feasibility, we have to look into the probabilistic constraints and distinguish the epistemic uncertainty and aleatoric uncertainty. First, when assuming the observation on the constraints are noise-free, namely $Y_{\mathcal{C}_k}(\mathbf{x}) = \mathcal{C}_k(\mathbf{x})$, we could simply use the indicator function $\mu_k$ for each constraint to turn the feasibility function into an indicator function.

$$r(\mathbf{x}) = \begin{cases} f(\mathbf{x}) & if \quad \mathbb{I}(C_k(\mathbf{x}) > h_k) \quad \forall k \in \mathbf{K} \\ -inf & o.w \end{cases} \tag{20}$$

Next, if the observation on the constraints is perturbed with a known Gaussian noise, namely $Y_{\mathcal{C}_k}(\mathbf{x}) \sim \mathcal{N}(\mathcal{C}_k(\mathbf{x}), \sigma)$, we could deal with the aleatoric uncertainty with a user-specific confidence level for each constraint $\mu_k \in (0, 1), \forall k \in \mathbf{K}$. Then we could turn $\mathbb{I}(Y_{C_k}(\mathbf{x}) > h_k)$ into probabilistic constraints following the definiation proposed by Gelbart et al. (2014) and

$$\mathbb{P}\left[Y_{C_k}(\mathbf{x}) > h_k\right] \geq \mu_k$$

to explicitly deal with the aleatoric uncertainty. With the percentage point function (PPF), we could transform the probabilistic constraints into a deterministic constraint $\mathbb{I}(C_k(\mathbf{x}) > \hat{h}_k)$ with $\hat{h}_k = PPF(h_k, \sigma, \mu_k)$, meaning $\hat{h}$ is the $\mu_k$ percent point of a Gaussian distribution with $h_k$ and $\sigma$ as its mean and standard deviation. Hence, we could unify the form of rewards of noise-free and noisy observation on the constraints with the user-specified confidence levels. For simplicity and without loss of generalization, we stick to the definition in equation 3 and let all $h_k = 0$.

Throughout the rest of the paper, we want to efficiently locate the global maximizer

$$\mathbf{x}^* = \underset{\mathbf{x} \in \mathbf{X}, \forall k \in \mathbf{K}, \mathcal{C}_k(\mathbf{x}) > 0}{\arg\max} f(\mathbf{x})$$

Equivalently, we seek to achieve the performance guarantee in terms of simple regret at certain time $t$,

$$\mathbf{R}_t := r(\mathbf{x}^*) - \underset{\mathbf{x} \in \{\mathbf{x}_1, \mathbf{x}_2, \ldots \mathbf{x}_t\}}{\max} r(\mathbf{x})$$

with a certain probability guarantee. Formally, given a certain confidence level $\delta$ and constant $\epsilon$, we want to guarantee that after using up certain budget $T$ dependent on $\delta$ and $\epsilon$, we could achieve a high probability upper bound of the simple regret on the identified area $\hat{\mathbf{X}}$ which is the subset of $\mathbf{X}$.

$$P(\underset{\mathbf{x} \in \hat{\mathbf{X}}}{\max} \mathbf{R}_T(\mathbf{x}) \geq \epsilon) \leq 1 - \delta$$

# D  DATASET

Here we offer a more detailed discussion over the construction of the six CBO tasks studied in section 6.

## D.1  SYNTHETIC TASKS

We study two synthetic CBO tasks constructed from conventional BO benchmark tasks. Here we rely on the implementation contained in BoTorch's (Balandat et al., 2020) test function module.

**Rastrigin-1D-1C**  The Rastrigin function is a non-convex function used as a performance test problem for optimization algorithms. It was first proposed by Rastrigin (1974) and used as a popular benchmark dataset (Pohlheim). It is constructed to be highly multimodal, with local optima being regularly distributed to trap optimization algorithms. Concretely, we negate the 1D Rastrigin function and try to find its maximum: $f(\mathbf{x}) = -10d - \sum_{i=1}^{d} \left( x_i^2 - 10\cos(2\pi x_i) \right), \ d = 1$. The range of $\mathbf{x}$ is $[-5, 5]$, and we construct the constraint to be $c(\mathbf{x}) = |\mathbf{x} + 0.7|^{1/2}$. When setting the threshold as $\sqrt{2}$, we essentially exclude the global optimum from the feasible area. The constraint enforces the optimization algorithm to explore feasibility rather than allowing algorithms to improve the reward by merely optimizing the objective. Then, the feasible region takes up approximately 60% of the search space. This one-dimensional task is designed to illustrate the necessity of adaptively trade-off learning of constraints and optimization of the objective.

We also vary the threshold to control the portion of the feasible region to study the robustness of COBALT. Figure 3 shows the distribution of the objective function and feasible regions on the samples.

**Ackley-5D-2C**  The Ackley function is also a popular benchmark for optimization algorithms. Compared with the Rastrigin function, it is similarly highly multimodal, while the region near the center is growingly steep. Same as what is done for Rastrigin, we negate the 5D Ackley function and try to find its maximum: $f(\mathbf{x}) = 20\exp\left(-0.2\sqrt{1/d\sum_i^d x_i^2}\right) + \exp\left(1/d\sum_i^d \cos(2\pi x_i)\right) + 20 + \exp(1), \ d = 5$. The search space is restricted to $[-5, 3]^5$. We construct two constraints to enforce a feasible area approximately taking up 14% of the search space. The first constraint $(\|x - \mathbf{1}\|_2 - 5.5)^2 - 1 > 0$ constructs two feasible regions with one in the center and the other close to the boundary of the search space. The second constraint $-\|x\|_\infty^2 + 9$ allows one hypercube feasible region in the center.

## D.2  REAL-WORLD TASKS

We study four real-world CBO tasks. The first three are extracted from Tanabe and Ishibuchi (2020), which offers a broad selection of real-world multi-objective multi-constraints optimization tasks. The fourth one is a 32-dimensional optimization task extracted from the UCI Machine Learning repository (mis, 2019).

**Vessel-4D-3C**  The pressure vessel design problem aims at optimizing the total cost of a cylindrical pressure vessel. The four variables represent the thicknesses of the shell, the head of a pressure vessel, the inner radius, and the length of the cylindrical section. The problem is originally studied in Kannan and Kramer (1994), and we follow the formulation in RE2-4-3 in Tanabe and Ishibuchi (2020). The feasible regions take up approximately 78% of the whole search space.

**Spring-3D-6C**  The coil compression spring design problem aims to optimize the volume of spring steel wire, which is used to manufacture the spring (Lampinen and Zelinka, 1999) under static loading. The three input variables denote the number of spring coils, the outside diameter of the spring, and the spring wire diameter, respectively. The constraints incorporate the mechanical characteristics of the spring in real-world applications. We follow the formulation in RE2-3-5 in Tanabe and Ishibuchi (2020). The feasible regions take up approximately 0.38% of the whole search space.

**Car-7D-8C**  The car cab design problem includes seven input variables and eight constraints. The problem is originally studied in Deb and Jain (2013). We follow the problem formulation in RE9-7-1 in Tanabe and Ishibuchi (2020) and focus on the objective of minimizing the weight of the car while meeting the European enhanced Vehicle-Safety Committee (EEVC) safety performance constraints. The seven variables indicate the thickness of different parts of the car. The feasible feasible region takes up approximately 13% of the whole search space.

**Converter-32D-3C**  This UCI dataset we use consists of positions and absorbed power outputs of wave energy converters (WECs) from the southern coast of Sydney. The applied converter model is a fully submerged three-tether converter called CETO. 16 WECs 2D-coordinates are placed and optimized in a size-constrained environment (mis, 2019). The input is, therefore, 32 dimensional. We place three constraints on the tasks, including the absorbed power of the first two converters being above a certain threshold of 96000 and the general position being not too distant with the two-norm below 2000. The feasible feasible region takes up approximately 27% of the whole search space.

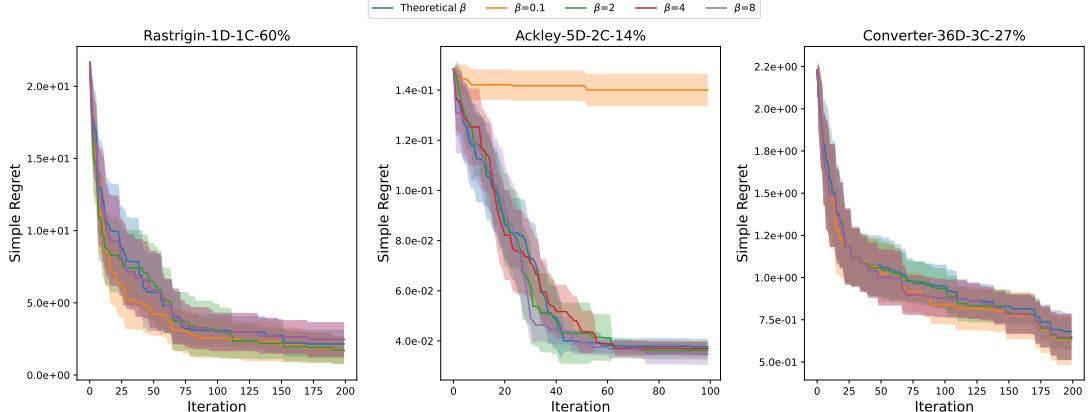

Figure 5: The figure illustrates the simple regret for a different choice of constant $\beta$ for COBALT. Here the theoretical $\beta$ are 6.51 for Rastrigin-1D-1C, 6.47 for Ackley-5D-2C, and 6.51 for Converter-36D-3C. The results are collected from 15 independent trials.

## E  ADDITIONAL EXPERIMENTS

Here we provide additional experiment results on COBALT.

### E.1  ROBUSTNESS TO CHOICES OF $\beta$

As is shown in figure 5, the algorithm is robust to moderate values of $\beta$. Except from the Ackley $\beta = 0.1$ where the filtering of ROI is over-aggressive and traps the model on a certain locality when a very small number of candidates remain in ROI. We observe that certain $\beta$ choices could be slightly better but don't impact the convergence and lack statistical significance. We believe the acquisitions in Eq. (6) and Eq. (5), together with the $\hat{X}$ identification when the models are well-fitted, contribute to this robustness. Different from conventional GP-UCB (Srinivas et al., 2009), the acquisition functions are standardized with the (maximum) lower confidence bound. The search domains are filtered when historical observations suggest poor performance in nearby areas.

### E.2  WALL TIME OF METHODS IN SECTION 6

We show the wall time of COBALT compared with the baselines in table 1. The results demonstrate the efficiency of COBALT due to the ROI filtering reducing the search space, though the ROI identification incurs additional cost for membership check.

| Problem | COBALT | CMES-IBO | SCBO | cEI |
|---|---|---|---|---|
| Rastrigin-1D-1C | 144.29 | 545.83 | 32.39 | 231.12 |
| Ackley-5D-2C | 96.19 | 565.10 | 25.43 | 180.39 |
| Converter-36D-3C | 190.05 | 660.27 | 31.73 | 267.36 |

Table 1: Average wall time (sec) of Different CBO Methods collected from 15 independent trials.

## F  ADDITIONAL EXPLANATION OF ALGORITHM 1

For algorithm 1, $\{x_{g_t,t}\}$ in line 11 are acquired in line 7 as $x_{\mathcal{C}_k,t}$ or line 9 as $x_{f,t}$, since $\mathcal{G}$ is composed of $\mathcal{C}_k$ and $f$. Roughly speaking, we are taking $\arg\max_{g,x}$, yet we avoid using such notation for two reasons. (1) the domain where equation 5 and equation 6 are maximized are different; (2) the domain for equation 6 could even be empty. Therefore, we are currently taking the $\arg\max$ of equation 5 and equation 6 over different domains (if not empty) separately and then taking the $\arg\max$ of the corresponding acquisition function values as in line 11.

## G  DISCUSSIONS

Here, we offer additional discussion over the concerns on COBALT.

### G.1  DIFFERENCE FROM EXISTING CBO METHOD WITH NO-REGRET GUARANTEE

We briefly discuss the essence of the differences from previous theoretical results in CBO. Lu and Paulson (2022) addresses equality constraints for instantaneous penalty-based regret. However, the reward formulation is quite different. Lu and Paulson (2023) offers theoretical results on cumulative regret and violations. Yet, they assume querying points out of the feasible region still yields reward and consider the violation separately.

**In general, we are not aware that their results lead to a similar guarantee as in our work when assuming querying infeasible point do not yield a reward. One key difference is that with the active learning component and feasibility assumption, we could guarantee to query a feasible point that bears a reward converging to optimal value with the desired confidence. In our specific reward formulation, we regard such a guarantee and, therefore, the contribution in algorithm design and analysis as sufficiently different from the previous work, even only considering the coupled setting.**

### G.2  EMPTY ROI(S)

It is possible that $\hat{\mathbf{X}}_t$ could be empty at certain $t$ when any intersection results in the empty set. However, according to the assumptions in section 5 and Lemma 1, the properly chosen $\beta_{f,t}$ and $\beta_{\mathcal{C},t}$ that does not result in over-aggressive filtering, the ROI is soundly defined. The algorithm is also robust to empty $U_{\mathcal{C}_k,t}$ due to the domain where the acquisition functions defined in equation 6 and equation 5 are maximized.

### G.3  COMPARABILITY

Despite both the acquisition function for optimization of the objective and active learning are confidence interval-based, it is possible they are not comparable. In practice, the objective and constraints could be of different scales. With prior knowledge of the scaling difference, one can choose to standardize the values or, equivalently, calibrate the acquisition function accordingly.

### G.4  LIMITATIONS

The limitation of COBALT includes (1) the inefficiency of identifying the ROIs due to the pointwise comparison in current implementation; (2) the lack of discussion over correlated unknowns, which are common in practice (e.g., two constraints are actually lower bound and upper bound of the same value). We expect the following work could further improve the algorithm's efficiency and effectiveness accordingly.

