# OpenReview forum: "Constrained Bayesian Optimization with Adaptive Active Learning of Unknown Constraints"
_ICLR.cc/2024/Conference — Submitted to ICLR 2024_

### Official Review · Reviewer_E2Xp · 2023-10-28

**Soundness:** 1 poor
**Presentation:** 2 fair
**Contribution:** 1 poor
**Rating:** 1
**Confidence:** 4

**Summary:**

The paper proposed a novel approach for combining active learning of unknown constraints with Bayesian optimization (CBO). The active learning criterion identifies regions of interest (ROI) based on confidence bounds on the output of the constraint function, and subsequently performs BO (UCB specifically) within the ROI. Theoretical analysis of the proposed algorithm is presented. Lastly, CBO is compared to other relevant algorithms, showing favorable results on synthetic and realistic tasks.

**Strengths:**

- __The methods section__ is clearly presented piece by piece, which makes it simple to digest.
- __Related works__ appear thorough, and cover relevant works from both the constrained BO and active learning litterature.
- __Results__ span from highly synthetic (aimed at showing traits of the algorithm rather than pure performance) to more realistic tasks.

**Weaknesses:**

To summarize, I believe this paper has substantial flaws both in terms of the novelty of the method, the theoretical analysis, and overall presentation. Unfortunately, this results in a paper that is far from publishable in its current state. Specifically:

- __The active learning component__ (the difference between the confidence bounds) is oddly presented and is not novel. The authors recognize this, but fail to recognize that the active learning method harks back to at least MacKay (1992) and has been widely applied since. Global variance reduction should not, in my opinion, be presented as a novel aspect of any algorithm. Moreover, it should be presented as Global variance reduction, or ALM (Active Learning MacKay), or otherwise, but _not_ as a difference between confidence bounds. This simply disguises the method as something seemingly more complex than it is.

- __Theoretical analysis__ contains substantial flaws, starting with Assumption 3:

  ___Assumption 3__ Given a proper choice of $\beta_t$ that is non-increasing, the confidence interval shrinks monotonically. (...)_.
This is flawed for two reasons:

  1. It appears trivial (beta is non-increasing, so the confidence interval will naturally be smaller as beta shrinks). _If there is a nuance to this assumption that I am missing, I encourage the authors to enlighten me._
  2. Assuming a non-increasing $\beta_t$ goes against a decade of BO convention, set by Srinivas et. al. (2009) that beta should scale as $\mathcal{O}(\log t)$.
Moreover, the subsequent Lemma 1 and Theorem have numerous un-introduced quantities ($K, \pi_t$, and appears to run contrary to Assumption 3. In Theorem 1,

 $\beta_t =2 \log( 2K+1)|D_{X_t}|$
where $D_{X_t}$ should be increasing in the data, making $\beta_t$ increase over time - thus not non-increasing. Admittedly, $D_{X_t}$ is the _intersection_ between a discretization of the search space and the data, which clearly contains only the data, at most. The non-increasing property of $\beta$ is re-stated in the Appendix, suggesting it is not a mistake.

The theoretical results are difficult to properly assess due to these apparent issues and ambiguities, so I will refrain from commenting on the correctness of the proofs in the appendix until these issues are clarified.

- __The algorithm__ _"Maximize the acquisition values from different aspects:"_ is informal, and line 11 is not well-defined. What does the $\text{argmax}$ over $\mathcal{G}$ (but not $x$) mean? Moreover, $x_{g, t}$ in line 12 has not been obtained from any prior step. Should the $g_t$ in line 11 really be $x_{g,t }$? As such, it is still not clear to me what the resulting acquisition function is, (but I believe it is the $\text{argmax}_{g, x}$ over all constraint and BO acquisitions, jointly.

- __The figures__ are difficult to parse due to the odd color choices (bright grey and dark orange on orange in Fig. 1, tiny legends everywhere).

- __Competing methods__ do not seem to be properly implemented. SCBO (Eriksson and Poloczek, 2019) builds on TuRBO, one of the more robust BO methods around. Seeing it fail on most tasks is a warning sign, and suggests to me that there are implementation flaws. If the authors believe it has been correctly implemented and used, I would suggest they motivate its poor performance in their specific setting.

__Minor__:
- Enumeration of contributions is generally a nice-to-have, and adds clarity.
- Hyperlinks are missing completely. Consider adding these.
- I would argue that the CBO acronym is occupied by Gardner et. al. (2014). While I don't believe there is a rule governing acronyms, I would strongly suggest for the authors to change it.


__Additional References__:

David MacKay. Information-based objective functions for active data selection. _Neural Computation_, 1992.

**Questions:**

In addition to the weaknesses (which certainly poses a few questions), I am curious about the following:

- The constraint functions $g$ are assumed to be continous-valued. Is this conventional, and is the method ammenable to non- continous (i.e. binary output) constraints?

**Details Of Ethics Concerns:**

None.

---

> ### Author Response · Authors · 2023-11-20
> **Response by Authors (1/3)**
>
> We sincerely appreciate the detailed comments by the reviewer and would like to respond in the following.
>
> ***Weakness***
>
> ```
> The active learning component is oddly presented and is not novel.
>
> ```
> We sincerely appreciate the reference pointed out by the reviewer, where the active data selection on the region of interest is thoroughly discussed. We acknowledge the similarities between our approach and the global variance reduction methods or ALM, as well as their roots in previous work. In the revision, we will clarify this connection and refrain from describing the active learning aspect of COBALT as entirely novel. However, we would like to highlight the difference between the active learning methods referred by the reviewer and the proposed algorithm for constrained BO tasks.
>
> We aim to propose a principled tradeoff of **active learning of multiple unknown constraints and the optimization of the objective in the feasible region**. To achieve this, we choose to unify the two different aspects---namely variance reduction and UCB, by framing them under the same mathematical framework as elaborated in Eq (5) and Eq (6). This framing ensures the comparability of the acquisition function of the unknown objective and the acquisition function of the unknown constraints. Further, it serves as the foundation for defining our acquisition function tailored to both needs. The other reason we use the interval between UCB and LCB is to follow the convention in previous works (AL-LSE[5] and BALLET[6]).
>
> Another significant difference between the proposed method and the global variance reduction is that we are doing **combined variance reduction** on adaptively identified **intersected ROI**, which leverages the UCB or LCB as the threshold.  The **shrinkage of the intersected ROI** guarantees the exploitation perspective of the acquisition and avoids budget waste, as discussed with reviewer AopU. The **variance reduction on ROI (as in eq (6)) and standardized GP-UCB** on ROI (as in eq (6)) achieve the exploration for both active learning and optimization. With the combination of ROI intersection and variance reduction on both the objective and the constraints, the proposed method achieves an **efficient and principled tradeoff** between exploration and exploitation. This serves the purpose of constrained BO and resonates with the well-known tradeoff in conventional BO.
>
> As pointed out by other reviewers, given the **specific constrained BO reward** discussed in this paper, this is the **first theoretical result of the convergence of the constrained BO** algorithm we are aware of.
>
> ```
> Theoretical analysis constraints substantial flaws: assumption 3 seems trivial.
> ```
> We’d like to point out that there is a subtle difference between a monotonically shrinking $2\beta_t^{1/2}(x)\sigma_{t-1}(x)$ and the consistency of confidence interval at different iterations, as discussed in assumption 3. The former is trivial given the monotonicity of $\\sigma_{t-1}(x)$, while the latter could possibly be violated even when $\beta_t$ is non-increasing. For example, if the observed $y_t$ actually lies on the confidence interval, the resulting posterior confidence interval at $x_t$ could disagree with the confidence interval of the prior. Assumption 3 actually assumes a consistency of prior and posterior confidence intervals, and it helps with the analysis by guaranteeing a consistently shrinking ROI.
>
> If we only resort to the monotonicity of the pointwise $ \sigma_{t-1}(x)$ when having a constant $\beta$ for $\forall 1 \geq t \geq T$ where T is the known total budget, we could revise the $\beta$ to be dependent only on $\tilde{D}$ rather than  $\tilde{D}_{\hat{X}_t}$. In that case, the theoretical results still hold. We refer to the detailed discussion in Appendix A of the revised paper.
>
> ```
> Non-increasing beta_t
> ```
>
> We appreciate the detailed question here. The reason $\beta_t=2\log(2(K+1)\vert \tilde{D}\vert \pi_t/ \delta)$ needs to grow is the need to guarantee the convergence of the series, which allows an asymptotical sublinear regret. Specifically, the requirement is $\sum_t{\pi_t} = 1$. Yet, in our problem setup, since cumulative regret could not be bounded as instantaneous regret could be infinite when the evaluated candidate $x_t$ violates any of the constraints (cf. Equation (3)), we aim to guarantee that after a certain horizon T, the *simple regret* could be bounded. Therefore, there is no need to have a convergence series $\sum_t{\pi_t} = 1$. As shown in the proof, we only need $\sum_{t=1}^T{\pi_t} = 1$ for a given budget/horizon T, and setting $\pi_t = 1/T$ satisfies it. We made revisions in the analysis section and the proof part to highlight this.

---

> > ### Author Response · Authors · 2023-11-20
> > **Response by Authors (2/3)**
> >
> > The setting in which $\sum_{t=1}^T{\pi_t} = 1$ is used instead of $\sum_t{\pi_t} = 1$ can be found in [7] as they also aim at a simple regret after a sufficiently large $T$. Similarly, in a multi-fidelity BO setting [8], the total budget is assumed to be given to attain a theoretical guarantee on the cumulative regret.
> >
> > ```
> > The algorithm "Maximize the acquisition values from different aspects:" is informal, and line 11 is not well-defined.
> > ```
> > We agree that the unconventional notation could be confusing and want to add additional clarification as the following. $\{x_{g,t}\}$ in line 11 are acquired in line 7 as $x_{\mathcal{C_k}, t}$ or line 9 as $x_{f, t}$, since $\mathcal{G}$ is composed of $\mathcal{C_k}$ and $f$. Roughly speaking, we are taking $arg\max_{g, x}$, yet we avoid using such notation for two reasons. (1) the domain where equation 5 and equation 6 are maximized are different; (2) the domain for equation 6 could even be empty. Therefore, we are currently taking the $arg\max$ of equation 5 and equation 6 over different domains (if not empty) separately and then taking the $arg\max$ of the corresponding acquisition function values as in line 11.
> >
> > ```
> > The figures are difficult to parse due to the odd color choices (bright grey and dark orange on orange in Fig. 1, tiny legends everywhere).
> > ```
> >
> > We sincerely appreciate the constructive comments and will revise the color and legends for easier parse.
> >
> > ```
> > Competing methods are not properly implemented. SCBO (Eriksson and Poloczek, 2019) builds on TuRBO, one of the more robust BO methods around.
> > ```
> >
> > We appreciate the concern raised by the reviewer. We’d like to point out that SCBO shows strong performance in the second and fifth columns of Figure 3 and in Car-Car-7D-8C in Figure 4.  We found its performance strongly depends on the initial center of the trust region. If the initial trust region is correctly identified, it demonstrates superior performance. The previous work [1] also shows a similar inconsistency of performance by TSC (a variant of SCBO) in Figure 3 of [1]. We would like to additionally cite the critics of SCBO by Takeno et al. (2022), that when the identified feasible region is empty, there is no rationale for selecting points of minimum constraint violations in the sense of TS.
> >
> > We appreciate the constructive comments by the reviewer and will accommodate additional discussion over SCBO performance in the revision.
> >
> > ```
> > Minors
> > ```
> >
> > We sincerely appreciate the constructive comments by the reviewer and would like to revise accordingly. Specifically, we enabled hyperlinks to the references in the revision. With regard to the CBO acronym, we appreciate the suggestion by the reviewer. We followed the recent practice of denoting the constrained BO tasks with CBO [1]. We will fix the caption of Figures 3 and 4 by replacing CBO with COBALT.
> >
> > Concerning the summarization of the contribution, we will improve the clarity through enumeration in the later revision. We want to highlight that we’ve listed the key contributions in section 7. We want to briefly reiterate that (1) We introduce COBALT to address constrained Bayesian optimization efficiently. The ROI intersection and acquisition function congregation allow the principled tradeoff between active learning of constraints and optimization of the objective. (2) The tradeoff allows the first convergence guarantee corresponding to the reward discussed in equation 3 we are aware of. (3) We offer comprehensive empirical evidence indicating the robustness and effectiveness of the proposed algorithm.

---

> > > ### Author Response · Authors · 2023-11-20
> > > **Response by Authors (3/3)**
> > >
> > > ***Question***
> > > ```
> > > In addition to the weaknesses (which certainly pose a few questions), I am curious about the following:
> > > The constraint functions are assumed to be continous-valued. Is this conventional, and is the method amenable to non-continuous (i.e., binary output) constraints?
> > > ```
> > >
> > > We follow the convention in constrained BO [1, 2] and active learning [4, 5] literature that assumes that the unknown constraint is sampled from a GP and there is the corresponding threshold defining the corresponding feasible area. The non-continuous constraints would require incorporating additional treatment, e.g., ROIAL, an ROI-based AL framework [3] that is designed for preferential or ordinal feedback. Their framework assumes that there is an underlying continuous utility function and uses information gain as the acquisition function, though no theoretical guarantee is provided.
> > >
> > > ***Reference***
> > >
> > > [1] Takeno, Shion, Tomoyuki Tamura, Kazuki Shitara, and Masayuki Karasuyama. "Sequential and parallel constrained max-value entropy search via information lower bound." In International Conference on Machine Learning, pp. 20960-20986. PMLR, 2022.
> > >
> > > [2] Gelbart, Michael A., Jasper Snoek, and Ryan P. Adams. "Bayesian optimization with unknown constraints." arXiv preprint arXiv:1403.5607 (2014).
> > >
> > > [3] K. Li et al., "ROIAL: Region of Interest Active Learning for Characterizing Exoskeleton Gait Preference Landscapes," 2021 IEEE International Conference on Robotics and Automation (ICRA), Xi'an, China, 2021, pp. 3212-3218, doi: 10.1109/ICRA48506.2021.9560840.
> > >
> > > [4] Bogunovic, Ilija, Jonathan Scarlett, Andreas Krause, and Volkan Cevher. "Truncated variance reduction: A unified approach to bayesian optimization and level-set estimation." Advances in neural information processing systems 29 (2016).
> > >
> > > [5] Gotovos, Alkis. "Active learning for level set estimation." Master's thesis, Eidgenössische Technische Hochschule Zürich, Department of Computer Science,, 2013.
> > >
> > > [6] Zhang, Fengxue, Jialin Song, James C. Bowden, Alexander Ladd, Yisong Yue, Thomas Desautels, and Yuxin Chen. "Learning Regions of Interest for Bayesian Optimization with Adaptive Level-Set Estimation." (2023).
> > >
> > > [7] Wang, Zi, and Stefanie Jegelka. "Max-value entropy search for efficient Bayesian optimization." In International Conference on Machine Learning, pp. 3627-3635. PMLR, 2017.
> > >
> > > [8] Song, Jialin, Yuxin Chen, and Yisong Yue. "A general framework for multi-fidelity bayesian optimization with gaussian processes." In The 22nd International Conference on Artificial Intelligence and Statistics, pp. 3158-3167. PMLR, 2019.

---

> > ### Comment · Reviewer_E2Xp · 2023-11-21
> > **Follow-up**
> >
> > __The active learning component is oddly presented and is not novel.__
> >
> > _We acknowledge the similarities between our approach and the global variance reduction methods or ALM, as well as their roots in previous work._
> >
> > I am not saying that there are similarities - I am saying that ALM/global variance reduction is clearly _equivalent_ to the proposed method (this is also shown in the paper's appendix). As such, I do not think the statement by the authors is fair.
> >
> > In the paper (Paragraph _Acquisition function for learning the constraints_) it is still not acknowledged that an ALM-equvialent strategy is used, despite it being proven in the appendix. This, to me, is the biggest issue with the paper, as I do not think it properly acknowledges previous work within the field. To this end, I think the formulation (i.e. as the difference between confidence bounds) disguises the fact that the method _already exists_. The two work cited for convention is published within the last year (and does not cite McKay, either) and the other does not seem to _evaluate the difference between bounds_ as the acquisition function. As such, I would not call this the convention.
> >
> > As I have several concerns regarding the theory of the paper, I will refrain from commenting further here.
> >
> > __Theoretical analysis constraints substantial flaws: assumption 3 seems trivial.__
> >
> > I do not see consistency mentioned in relation to that Assumption. Can the authors clarify further?
> >
> > __Non-increasing beta_t__
> >
> > Looking at the paper, it is still difficult to assess if $\beta_t$ is increasing or decreasing. Lemma 1 has  $\beta_t$ be a function of $\pi_t$  and Theorem 1 has  $\beta_t$ not be a function of $t$ at all, but of $T$.

---

> > > ### Comment · Reviewer_E2Xp · 2023-11-21
> > > **Follow-up, p2**
> > >
> > > __The algorithm "Maximize the acquisition values from different aspects:" is informal, and line 11 is not well-defined.__
> > >
> > > I would still encourage the authors to be more precise. _Maximize the acquisition values from different aspects_ is not precise enough to include in an algorithm, as the uninformed reader cannot devise what the algorithm does. Moreover, I think the authors need to expand on this specific line in the text.
> > >
> > > Lastly, my question regarding the intersection between the data and the discretization has yet to be addressed.

---

> ### Author Response · Authors · 2023-11-22
> **Response by Authors**
>
> Thank you for the swift response! We’d like to clarify the novelty further and answer the remaining question by the reviewer.
> ```
> I am saying that ALM/global variance reduction is clearly equivalent to the proposed method.
> ```
> Thank you for your constructive comments! We want to add the following clarification.
>
> We agree with the reviewer that the variance maximization defined in equation (6) exactly matches the discussion about maximum information gain in the ALM paper.  In fact, equation (6) itself is an AL-LSE [1] acquisition function, and equation (5) is a slightly modified UCB acquisition function. And we don’t intend to claim novelty on either the UCB or the variance reduction part. Please note that in the previous revision, we admitted the root of the active learning part in ALM in paragraph 3 of section 2. **We additionally clearly stated this inheritance in the latest revision in section 4.3.** However, the domain the acquisition function is maximized on is adaptively identified in a way that is not the same as the ALM discussion.
>
> We still want to reiterate our contribution here. The adaptive tradeoff of AL-LSE and GP-UCB to achieve an efficient constrained BO method that comes with theoretical justification on its convergence rate, given the specific reward form we discussed in equation (3), is non-trivial. As far as we are aware, the ALM paper and recent advances in CBO do not offer efficient, principled adaptive trade-offs between multiple acquisitions through the ROI intersection and acquisition function combination, as proposed in our work. We are not aware of the existing convergence guarantee to the reward corresponding to equation (3). Therefore, the proposed intersected ROI ($\hat{X}_t$ as defined in equation (4))-based method on top of the ALM differentiates it from the original ALM method, though the ALM paper uses ROI for different notions.
>
> In addition, please note the acquisition function for AL-LSE[1] is defined as the intersection of confidence intervals: $\vert\cap_{i=1}^t Q_i \vert = \vert \max_{i=1,..,t} LCB_i, \min_{i=1,...,t}UCB_i\vert$ and is maximized on an adaptively identified set.
>
> ```
> I do not see consistency mentioned in relation to that Assumption. Can the authors clarify further?
> ```
>
> Sure! The notion of consistency is stated in assumption 3, $UCB_{t_1}(x) \geq UCB_{t_2}(x)$ and $LCB_{t_1}(x) \leq LCB_{t_2}(x)$. While the monotonical shrinking that naturally holds due to the monotonicity of $\sigma_t$ only guarantees that $UCB_{t_2}(x)-LCB_{t_2}(x) \leq UCB_{t_1}(x)-LCB_{t_1}(x)$, when $t_1 \leq t_2$. We wanted to highlight that the consistency doesn’t naturally hold, in contrast to the shrinkage of the pointwise confidence interval.
>
>
> ```
> Looking at the paper, it is still difficult to assess if $\beta_t$ is increasing or decreasing. Lemma 1 has $\beta_t$ be a function of $\pi_t$, and Theorem 1 has $\beta_t$ not be a function of $t$ at all, but of $T$.
> ```
>
> Please note that in addition to the previous clarification where we justify that, in $\beta_t = 2log(2(K+1)\vert \tilde{D} \vert \pi_t / \delta)$ could be non-increasing as long as we don’t want a convergent series $\sum_t\pi_t^{-1} = 1$ but instead we only need $\sum_{t=1}^T\pi_t ^{-1}= 1$, we set $\pi_t = T$ as $T$ is the preset constant horizon meeting requirement in theorem 1. Then $\beta=2log(\frac{2(K+1)\vert \tilde{D} \vert T}{  \delta})$. Here the number of constraints $K$, the size of the discretization $\vert \tilde{D} \vert$, the horizon $T$, and $\delta$ are all given numbers. Then, $\beta$ is a constant.
>
>
> ```
> I would still encourage the authors to be more precise. Maximizing the acquisition values from different aspects is not precise enough to include in an algorithm, as the uninformed reader cannot devise what the algorithm does. Moreover, I think the authors need to expand on this specific line in the text.
> ```
> We sincerely appreciate these constructive comments. We added additional discussion to clarify it in the revision.
>
> ```
> Lastly, my question regarding the intersection between the data and the discretization has yet to be addressed.
> ```
> We apologize for missing it in the previous response. In fact, we believe $\tilde{D}$ as defined in Lemma 1 is a preset discretization in contrast to the $D_t$, which is the growing collection of historical observations. And $\hat{X}_t$ is the subset of the search space $\mathbf{X}$. Therefore, we do not intend to describe $\tilde{D}\_{\hat{X}_t}$ as the intersection of discretization and data but the discretization of the subset of search space. And as described in the previous response, we make $\beta=2log(\frac{2(K+1)\vert \tilde{D} \vert}{ T \delta})$ instead of relying on $\tilde{D}\_{\hat{X}_t}$ now.
>
> ***reference***
>
> [1] Gotovos, Alkis. "Active learning for level set estimation." Master's thesis, Eidgenössische Technische Hochschule Zürich, Department of Computer Science,, 2013.

---

### Official Review · Reviewer_AopU · 2023-10-30

**Soundness:** 2 fair
**Presentation:** 2 fair
**Contribution:** 2 fair
**Rating:** 3
**Confidence:** 4

**Summary:**

This study presents a solution to the constrained Bayesian optimization problem with theoretical analysis. The method involves defining high-confidence regions of interest (ROIs) for both the objective and constraints. Following this, an acquisition function is constructed to simultaneously optimize the objective and identify feasible regions. The effectiveness of the proposed algorithm is showcased through numerous experiments.

**Strengths:**

It is uncommon to find a paper on constrained Bayesian Optimization that includes theoretical analysis. This paper explores both coupled and decoupled settings, with the decoupled setting detailed in the appendix.

**Weaknesses:**

Unfortunately, the paper lacks clarity and precision, with several confusing and potentially erroneous arguments.

1. The high-level approach is inefficient for the following reasons:

   a. $x_t$ can be picked as $x\_{\\mathcal{C}\_k,t} = \\text{arg}\\!\\max\_{x \\in U\_{\\mathcal{C}\_k,t}} \\alpha\_{\\mathcal{C}\_k,t}(x)$, so it may belong to $L_{\mathcal{C}_{k'},t}$ for some $k' \neq k$. In this case, it raises the question of why the objective and constraints are evaluated at an infeasible input.

   b. In equation (5), when $\text{LCB}_{f,t,\max} = \infty$, we basically pick the most uncertain $f(x)$. However, there may existing uncertain $f(x)$ with a very low evaluation. Then, why do we need to pick this input if we are interested in large $f(x)$?

2. Here are some arguments that might be incorrect:

   a. The paper asserts that existing constrained Bayesian optimization lacks theoretical analysis. However, the work titled "No-Regret Bayesian Optimization with Unknown Equality and Inequality Constraints using Exact Penalty Functions" by Lu and Paulson (2022) provides theoretical analysis.

   b. The value of $\beta_t$ is defined based on $\hat{X}_t$ but $\hat{X}_t$ is defined based on $\beta_t$. Therefore, $\beta_t$ cannot be defined in Theorem 1.

   c. In the proof of Theorem 1 (proof of Lemma A.1): Lemma 5.4 of Srinivas et al. (2009) is not applied correctly. Lemma 5.4 is intended for *points selected* in Srinivas et al. (2009). However, in this paper, it is applied to the sequence of $x_{g,t}$ for all $g \in \mathcal{G}$ and $t$. It is important to note that not all $x_{g,t}$ are *selected points*: At each iteration only 1 input $x_{g,t}$ of a function $g \in \mathcal{G}$ is picked as the candidate to evaluate.

3. There are several unclear points:

   a. The paper may need more arguments to explain why the approach of Takeno et al. (2022) "violates the theoretical soundness".

   b. Theorem 1 does not address feasibility. Furthermore, Srinivas et al. (2009) analyze the regret of *some input* (for example $x_t$). While this paper introduces a definition of regret (or reward) of inputs, it does not use this definition in the theoretical analysis. This raises questions about the significance of Theorem 1 in terms of the algorithm's achieved reward (or regret).

   c. The abstract highlights a setting where "the objective and constraints can be independently evaluated", yet the paper primarily focuses on the coupled setting where they are evaluated together. The decoupled setting is briefly explained in the appendix without any supporting experiments.

   d. The paper introduces $\epsilon_C$ for theoretical analysis but does not utilize it in the algorithm. If the algorithm is used with $\epsilon_C = 0$, the theoretical analysis loses its significance as $T \ge \infty$.

   e. The abstract mentions a discussion on "the fundamental challenge of deriving practical regret bounds for CBO algorithms", but this discussion cannot be located in the paper."


4. There are other issues such as

   a. The chosen performance metric is simple regret. However, in practical applications, identifying the input with minimum regret is unknown. Thus, the paper does not address what input Bayesian Optimization recommends as the optimal solution.

   b. The experiment section is quite short. Surprisingly, in the Rastrigin-1D-1C, the discrete search space consists of $1000$ points but the proposed algorithm reaches the optimum with $2000$ iterations. Furthermore the noise is small $\mathcal{N}(0,0.1)$ compared to the function range $[-40,0]$. This efficiency is questionable.

   c. Assumption 2 does not hold for equality constraints.

5. There are some typos in the paper:

   a. In equation (5), $\infty$ should be replaced with $-\infty$.

   b. Before equation (1), GP\supscript(()

   c. In the appendix, there are several typos: thoeretical, defination, Cauchy-Schwar.

   d. Both $\epsilon_k$ and $\epsilon_C$ refer to the same thing.

   e. The label of the proposed algorithm in Figures 3 and 4 should be COBALT instead of CBO.

   f. It's worth noting that the name COBALT coincides with an existing Bayesian Optimization work titled 'COBALT: COnstrained Bayesian optimizAtion of computationaLly expensive grey-box models exploiting derivaTive information' by Paulson and Lu (2021).

**Questions:**

Please answer to the aforementioned weaknesses.

---

> ### Author Response · Authors · 2023-11-20
> **Response by Authors (1/2)**
>
> We appreciate the detailed and constructive comments by the reviewer. We’d love to clarify the following.
>
> ***Weakness***
>
> ***1. The high-level approach is inefficient for the following reasons:***
>
> ```
> a. Querying $L_{C_{k’}}$ for some $k’ \neq k$.
> ```
>
> Yes, this motivates the design of the intersection of ROIs and different ROIs for objectives and constraints. We want to avoid inefficient queries. This is a very good point. Actually, the reviewer might refer to the code we provided (src/opt/constrained.py line 523-524); we are actually intersecting $U_{C_k,t}$ with $\hat{X_t} $ instead of maximizing on the full $U_{C_k,t}$ on line 7 in algorithm 1. This is actually a typo, and we apologize for the confusion. Also, please note that in the proof of theorem 1, we’ve been using the correct domain.
>
> ```
> b. In equation (5), pick the most uncertain f(x).
> ```
>
> Please note that we are picking the most uncertain $f(x)$ on ROI, where the $UCB_{f} (x)$ is guaranteed to be above a certain threshold. Intuitively, we are narrowing the confidence interval of $f(x^*)$. This serves as an efficient global optimization acquisition as the uncertainty maximization guarantees the exploration while the ROI shrinking guarantees the exploitation. When $LCB_{f,t,max}$ is infinity, it means there is no super level-set identified, and taking the maximum of UCB could be optimizing the objective function out of the feasible region, which doesn’t always contribute to the performance.
>
>
> ***2. The arguments that might be incorrect.***
>
> ```
> a. With respect to the claim that Constrained Bayesian optimization lacks theoretical analysis, the work by Lu and Paulson provides theoretical analysis.
> ```
>
> We’re grateful for the reference provided by the reviewer and would like to add additional discussion to the related work.  We would like to highlight that regret is different, and we provide theoretical justification in the task following the constrained BO problem setup where the infeasible candidate doesn’t incur a reward.
>
> ```
> b. $\beta_t$ cannot be defined in theorem 1.
> ```
> We appreciate the reviewer pointing out the typo here. Actually, the reason we introduce assumption is to guarantee a monotonically shrinking $\hat{X_t}$, meaning we could use $\hat{X}\_{t-1}$ as the conservative surrogate for $\hat{X_{t}}$. In the revision, we will avoid confusion by using $\tilde{D}$ instead.
>
> ```
> c. In the proof of theorem 1 (proof Lemma A.1), Lemma 5.4 of Srinivas is not applied correctly applied.
> ```
> We appreciate the reviewer pointing out the typo here. We correct the proof in the revision by replacing $x_{g,t}$ with $x_t$. Please note that the ultimate acquisition function is defined as the maximum of the multiple acquisition functions. Roughly speaking, though only one of the $x_{g, t}$  is picked as $x_t$, the upper bound holds for the other points that have not been picked, as is shown in the revised proof in section A.2.
>
> ***3. There are several unclear points.***
> ```
> The paper may need more arguments to explain why the approach of Takeno et al. (2022) "violates the theoretical soundness."
> ```
> We acknowledge that the criticism could be overstated. Yet, as admitted in Takeno et al. (2022), there is no regret guarantee for the entropy-based method. It proves that the link to the UCB method in previous work could be an error.
>
> ```
> b. Theorem 1 does not address the feasibility. Furthermore, Srinivas et al. (2009) analyze the regret of some input. While this paper introduces a definition of regret (or reward) of inputs, it does not use this definition in the theoretical analysis. This raises questions about the significance of Theorem 1 in terms of the algorithm's achieved reward (or regret).
> ```
> (1) In assumption 2, we assume the feasibility. Theorem 1 shows that if there are feasible points, we can identify them after sufficient query with a certain confidence. Otherwise, the empty super level-set would suggest that with a certain confidence, assumption 2 is violated. That is, there is no feasible area.
>
> (2) With regards to actual reward, if the identified Superlevel set is non-empty, the acquisition function $acq_{f,t}$ on that would actually be $UCB_f(x)$, which shares the regret guarantee with Srinivas et al. (2009). If the user wants to guarantee a query on the argmax of $UCB_f(x)$, the user can enforce that at any time or afterward. Otherwise, the algorithm is likely to query at least the argmax of $UCB_f(x)$ once within a large interval. We also add an additional corollary of theorem 1 in the revision to show the guarantee of a simple regret.

---

> > ### Author Response · Authors · 2023-11-20
> > **Response by Authors (2/2)**
> >
> > ```
> > c. The abstract highlights a setting where "the objective and constraints can be independently evaluated," yet the paper primarily focuses on the coupled setting where they are evaluated together. The decoupled setting is briefly explained in the appendix without any supporting experiments.
> > ```
> >
> > We do provide the algorithm and analysis as an interesting side product for the decoupled setting. Yet, we do not intend to highlight the focus on the decoupled setting due to the limited scope of this work and motivating application. We would leave the comprehensive study of decoupled settings into future work. We would tune down the sentence in the revision.
> >
> > ```
> > d. The $\epsilon_C=0$ makes the analysis lose its significance.
> > ```
> > We believe this is a critical revelation of our analysis. That is, when the global optimum lies on the exact boundary defined by one of the constraints, due to the nature of active learning under noise, there is no guarantee that the algorithm can always identify the optimum points on the boundary with the desired confidence fixed beforehand.
> >
> > ```
> > e. The fundamental challenge of deriving practical regret bounds for CBO algorithms.
> > ```
> >
> > We apologize for the confusion here. As is discussed above, 3.d and revealed in assumption 2. If we lack the knowledge of $\epsilon_c$, the current AL-LSE algorithm doesn't guarantee the ability to identify the entire boundary with the desired confidence. In the worst case, if the feasible area is only on the boundary as $\epsilon_c \rightarrow 0$, there is no guarantee that with the desired preset confidence, we could evaluate within the feasible area and hence incur a regret that could be bounded.
> >
> > ***4. There are other issues, such as.***
> >
> > ```
> > a. The chosen performance metric is simple regret. However, identifying the input with minimum regret is unknown in practical applications. Thus, the paper does not address what input Bayesian Optimization recommends as the optimal solution.
> > ```
> > We appreciate the comment and add additional discussion over the guarantee of a simple regret by picking the $arg\max_{\hat{X_T}} LCB_{f, T}(x)$ in the revised appendix.
> >
> > ```
> > b. Surprisingly, in the Rastrigin-1D-1C, the discrete search space consists of 1000 points, but the proposed algorithm reaches the optimum with 2000 iterations. Furthermore, the noise is small (0,0.1) compared to the function range [−40,0]. This efficiency is questionable.
> > ```
> >
> > We agree that on conventional BO tasks on the 1D Rastrigin without unknown constraints, the 2000 iteration seems overkill, especially given the small noise. Yet the challenge is poised by the tradeoff between learning the feasibility and optimizing the objective with noisy observation. The performance of the baselines after 2000 iterations reveals that the problem lacks a principled tradeoff. We consider a small perturbation on the constraint significant due to the distribution of Rastrigin-1D and the positions of the optimal area. Additionally, the simple regret actually drops fast in the beginning 400 iterations. It is expected that the algorithm could query suboptimal several times due to noise and the existence of multiple identical objective values.
> >
> > ```
> > c. Assumption 2 does not hold for equality constraints.
> > ```
> > Yes, we believe this relates to the discussion in 3.d and 3.e., if the objective doesn't incur reward out of the feasible region, it is challenging to guarantee the performance without the knowledge of epsilon or when epsilon converges to 0. However, in practice, the user might slightly adjust the threshold $h_k$ to accommodate the $\epsilon_k$.
> >
> > ***5. Others***
> >
> > We sincerely appreciate the comments by the reviewer and will fix them correspondingly in the revision. We are grateful the reviewer pointed out the naming coincidence. We acknowledge this in the revision.

---

> > > ### Comment · Reviewer_AopU · 2023-11-21
> > > **Thanks for Your Reply and Clarification Needed on Certain Aspects**
> > >
> > > Thank you for the response, which gives me a clearer understanding of 1a, 1b, 2a, 3a, 4b, 4c. However, I am still facing difficulties in understanding some aspects of the responses. Furthermore, a review of the notations in the paper may be necessary to ensure consistency (this notation revision is for future revision following the rebuttal, not an immediate request).
> > >
> > > (2b) Could the authors provide a concise explanation for the need to ensure a monotonically shrinking $\hat{X}_t$ and elaborate on how defining $\beta_t$ using $\tilde{D}$ still ensures this property.
> > >
> > > (2c) My understanding of the application of Lemma 5.4 remains unclear. Suppose $x_{g,t}$ is not selected as $x_t$ for iterations from 1 to 5 for a function $g$; how can we demonstrate that $\sum_{t=1}^5 (2 \beta^{1/2} \sigma_{g,t-1}(x_t))^2 \le C_1 \beta \gamma_{g,5}$?
> > >
> > > (3b) I kind of understand (1) but I find (2) difficult to understand. Why "if the superlevel set is non-empty, ..., which shares the regret guarantee with Srinivas et al. (2009)"? Could you clarify what is meant by the superlevel set in this context? If it is either a strict superset or a strict subset of the feasible region, then could you provide the reasons why the regret is comparable with Srinivas et al. (2009) where the feasible region is X?
> > >
> > > (4a) Is the authors referring to Corollary 2 in the Appendix? This section lacks clarity, as it appears that $R_T$ is defined as $f^* - f(x_T)$ in this proof, which is not the same as the simple regret defined at the end of section 3. The use of both $t$ and $T$ in the same term within this proof seems to be typos and causes confusions.
> > >
> > > Concerning item 3c, leaving the comprehensive study of decoupled settings for future research could potentially lessen the impact of the paper. This is due to the fact that Lu and Paulson have previously conducted theoretical work on CBO, specifically addressing equality constraints which is not addressed in this paper.
> > >
> > > In the proof of Theorem 2, where does $g$ come from? This proof needs to be proofread and elaborated carefully.
> > >
> > >
> > > Minor points:
> > >
> > > (3d) There might be a misunderstanding of my question. I am specifically asking about "The paper introduces $\epsilon_C$ for theoretical analysis but does not utilize it in the algorithm." It means why $\epsilon_C$ is absent from the algorithmic description and only surfaces in the theoretical analysis (with no prior mention before section 5). If the algorithm sets $\epsilon_C = 0$, it implies that the analysis loses its significance for the proposed algorithm.
> > > This observation suggests ensuring consistency between the algorithmic description and the theoretical analysis with respect to $\epsilon_C$.
> > >
> > > (3e) I am expecting the paper would discuss a fundamental challenge of deriving practical regret bounds for CBO algorithms, which prevents existing works from deriving a theoretical analysis. Then, the paper will propose an algorithm to resolve it. However, from the response, it seems to be a limitation that is not resolved in this work.
> > >
> > > In lines 12 and 13 of Algorithm 2, the subscripts should be $g_t$ instead of $g,t$.

---

> ### Author Response · Authors · 2023-11-22
> **Clarification by Authors (1/3)**
>
> We are glad that we have addressed some of the reviewer’s concerns and would like to clarify the following further.
>
> ```
> (2b) Could the authors provide a concise explanation for the need to ensure a monotonically shrinking $\hat{X}_t$ and elaborate on how defining $\beta_t$ using $\tilde{D}$ still ensures this property?
> ```
> Sure! As noted by the reviewer in the original 2.b comments, at $t$, we can not actually estimate $\beta_t =2\log(\frac{2(K+1)\vert \tilde{D} \cap \hat{X}\_t \vert T}{\delta})$ because $\hat{X}\_t$ relies on the $\beta_t$. However, if $\hat{X}\_t$ is monotonically shrinking, meaning $\vert \tilde{D} \cap \hat{X}\_t \vert \leq  \vert\tilde{D} \cap \hat{X}\_{t-1}\vert $, we could define $\beta_t =2\log(\frac{2(K+1)\vert \tilde{D} \cap \hat{X}\_{t-1} \vert T}{\delta}) \geq 2\log(\frac{2(K+1)\vert \tilde{D} \cap \hat{X}\_t \vert T}{\delta})$, then the union bound taken in the proof of lemma 1 still hold.
>
> Now, we avoid the confusion by simply letting $\beta_t =2\log(\frac{2(K+1)\vert \tilde{D} \_t \vert T}{\delta}) \geq 2\log(\frac{2(K+1)\vert \tilde{D} \cap \hat{X}\_{t-1} \vert T}{\delta})$ as in the revised theorem 1. Then $\beta_t$ is essentially a constant. Though it does not guarantee that $\hat{X}\_t$ is monotonically shrinking, the desired union bound in lemma 1 still holds. The loss here is that $\beta_t$ could have been smaller if being defined upon $\vert\tilde{D} \cap \hat{X}\_{t-1}\vert$ instead.
>
> ```
> (2c) My understanding of the application of Lemma 5.4 remains unclear. Suppose $x_{g,t}$ is not selected as $x_t$ for iterations from 1 to 5 for a function $g$; how can we demonstrate that $\sum_{t=1}^5 (2 \beta^{1/2} \sigma_{g,t-1}(x_t))^2 \le C_1 \beta \gamma_{g,5}$?
> ```
>
> We’d like to address the reviewer’s concern but are not sure if we fully understand the question here. In the original 2.c comments, the reviewer pointed out that Lemma 5.4 only applies to the actually selected sequence of $x_t$. We believe here $\sum_{t=1}^5 (2 \beta^{1/2} \sigma_{g,t-1}(x_t))^2 \le C_1 \beta \gamma_{g,5}$ exactly match Lemma 5.4. Please note that in the proof of lemma 5.4, the sum of the square of simple regret $\sum_t r^2_t$ is first transformed into $\sum_{t=1} (2 \beta^{1/2} \sigma_{t-1}(x_t))^2$ by lemma 5.1 and 5.2 of Srinivas et al. (2009).
>
> In addition, we want to highlight that the bridge between $2 \beta^{1/2} \sigma_{g,t-1}(x_{g,t})$ and the usage of lemma 5.4  is $\sum_{t=1}^{T} \alpha_t^2 \leq \sum_{t=1}^{T} \sum_{g \in \{C_k\}\_{k\in\mathcal{K}} } (\alpha_{g,t}(x_{t}))^2$, which is currently in the proof of our lemma A.1. For the left-hand side we have $2 \beta^{1/2} \sigma_{g,t-1}(x_{g,t}) \leq \alpha_t$ in the scenarios we’ve discussed in formula (8)-(12) in the proof of lemma A.1. And for the right-hand side we have $\alpha_{g,t}(x_{t}) \leq 2 \beta^{1/2} \sigma_{g,t-1}(x_t)$ by definition of the acquisition functions $\alpha_{g,t}$.
>
> ```
> (3b) I kind of understand (1) but I find (2) difficult to understand. Why "if the superlevel set is non-empty, ..., which shares the regret guarantee with Srinivas et al. (2009)"? Could you clarify what is meant by the superlevel set in this context? If it is either a strict superset or a strict subset of the feasible region, then could you provide the reasons why the regret is comparable with Srinivas et al. (2009) where the feasible region is X?
> ```
> We apologize for the confusion here. First, the superlevel set we refer to in this context is $S_{C,t}$, which is defined in section 4.2. In practice, since we are working on the discretization, it should be $\tilde{D} \cap S_{C,t}$ in the implementation. As illustrated in equation (5) in section 4.3, when $S_{C,t}\neq \emptyset$, $\alpha_{f,t}$ is essentially $UCB_{f,t}$. Yet we find out that the originally claimed connection to the regret guarantee with Srinivas et al. (2009) might be indirect, as the results by Srinivas et al. (2009) are on cumulative regret when using UCB as an acquisition function, while we are aiming at a simple regret guarantee.
>
> We would like to refer to the updated Corollary 2, where we now show that when picking the point maximizing $UCB_{f,t}$ on $\tilde{D} \cap S_{C,t}$, we could translate the theorem 1 into a simple regret bound.

---

> > ### Author Response · Authors · 2023-11-22
> > **Clarification by Authors (2/3)**
> >
> > ```
> > (4a) Is the authors referring to Corollary 2 in the Appendix? This section lacks clarity, as it appears that $R_T$ is defined as $f^* - f(x_T)$ in this proof, which is not the same as the simple regret defined at the end of section 3. The use of both $t$ and $T$ in the same term within this proof seems to be typos and causes confusion.
> > ```
> > Yes, we refer to Corollary 2. To better answer (3b), we updated the corollary 2. Please note that $R_T$ is not defined as $f^* - f(x_T)$ but still follows the definition of simple regret. The resulting bound of $f^*-f(x_T) < \epsilon$ and the fact $x_T \in \tilde{D}\_{\hat{X_T}} \cap S_{C,t}$ makes $R_T \leq  f^*-f(x_T) < \epsilon$ hold for probability at least $1-\delta$. We’ve made the update to reflect that. We also improved the clarity by replacing $t$ with $T$ in the latest revision.
> >
> > ```
> > Concerning item 3c, leaving the comprehensive study of decoupled settings for future research could potentially lessen the impact of the paper. This is due to the fact that Lu and Paulson have previously conducted theoretical work on CBO, specifically addressing equality constraints, which are not addressed in this paper.
> > ```
> >
> > We agree with the reviewer that Lu and Paulson’s previous theoretical results in [1] address equality constraints for instantaneous penalty-based regret. However, the reward formulation is quite different. During the rebuttal period, we also noticed another recent pre-print work by the same authors offering theoretical results on cumulative regret and violations [2]. Yet, they assume querying points out of the feasible region still yields reward and consider the violation separately.
> >
> > In general, we are not aware that their results lead to a similar guarantee as in our work when assuming querying infeasible point do not yield a reward. One key difference is that with the active learning component and feasibility assumption, we could guarantee to query a feasible point that bears a reward converging to optimal value with the desired confidence.  In our specific reward formulation, we regard such a guarantee and, therefore, the contribution in algorithm design and analysis as sufficiently different from the previous work, even only considering the coupled setting. At the same time, we acknowledge the solid contribution of the previous works [1, 2], where different rewards lead to different algorithm designs. We’ve added a corresponding discussion in section 2.
> >
> > ```
> > In the proof of Theorem 2, where does $g$ come from? This proof needs to be proofread and elaborated carefully.
> > ```
> > We appreciate the constructive comments by the reviewer and have extended the proof to be more self-contained. We are not sure which one the reviewer is pointing to, but we believe $g_t$ should be taken according to algorithm 2.

---

> ### Author Response · Authors · 2023-11-22
> **Clarification by Authors (3/3)**
>
> ***Minor points:***
> ```
> (3d) There might be a misunderstanding of my question. I am specifically asking about "The paper introduces $\epsilon_C$ for theoretical analysis but does not utilize it in the algorithm." It means why $\epsilon_C$ is absent from the algorithmic description and only surfaces in the theoretical analysis (with no prior mention before section 5). If the algorithm sets $\epsilon_C = 0$, it implies that the analysis loses its significance for the proposed algorithm.
> This observation suggests ensuring consistency between the algorithmic description and the theoretical analysis with respect to $\epsilon_C$.
> ```
> We appreciate the clarification by the reviewer, and we would address this inconsistency by explicitly including the estimation of $\epsilon_C$ as an input of the algorithm.
>
> ```
> (3e) I am expecting the paper to discuss a fundamental challenge of deriving practical regret bounds for CBO algorithms, which prevents existing works from deriving a theoretical analysis. Then, the paper will propose an algorithm to resolve it. However, from the response, it seems to be a limitation that is not resolved in this work.
> ```
> We believe it is fundamental because we regard the limitation that the active learning component can not identify the full boundary of a dense search space with confidence within the limited number of queries as fundamental. Yet, we agree with the reviewer that there is no proof that the problem of active learning is a necessity, even considering our definition of reward.
>
> ```
> In lines 12 and 13 of Algorithm 2, the subscripts should be $g_t$ instead of $g,t$.
> ```
> We are grateful for the correction and have revised it accordingly. We found it actually should be corrected to be $g_{t}, t$ instead of $g_t$. Here, $g_t$ denotes the function to be evaluated at $t$, while the other $t$ is needed to ensure the alignment with subscript $C_k, t$ in line 7 and  $f, t$ in line 9.
>
>
> ***Reference***
>
> [1] Lu, Congwen, and Joel A. Paulson. "No-regret Bayesian optimization with unknown equality and inequality constraints using exact penalty functions." IFAC-PapersOnLine 55, no. 7 (2022): 895-902.
>
> [2] Lu, Congwen, and Joel A. Paulson. "No-regret constrained Bayesian optimization of noisy and expensive hybrid models using differentiable quantile function approximations." arXiv preprint arXiv:2305.03824 (2023).

---

> > ### Comment · Reviewer_AopU · 2023-11-22
> >
> > Thank you for providing the updated paper and addressing my questions. After reviewing the response, there is a remaining issue related to (3b) and (4a). The proof of the simple regret in Corollary 2 is "after attaining 13". But the definition $\beta = 1/T$ is established based on the iteration T when 13 is attained. Consequently, "after attaining 13", the upper confidence bound and lower confidence bound (depending on $\beta$) may no longer be applicable. So, this is potentially an error in the proof of the simple regret.
> >
> > The following are just some comments and clarifications.
> >
> > (3c) The comparison with existing works may be improved by including a discussion on the difference in the assumptions.
> >
> > (2c) My question is from the previous response
> > ```
> > Roughly speaking, though only one of the $x_{g,t}$ is picked as $x_t$, the upper bound holds for the other points that have not been picked, as is shown in the revised proof in section A.2.
> > ```
> > My question is that assuming $x_{g',t}$ is not selected as $x_t$ (for a function $g'$), how do we show that the upper bound holds for these "other points that have not been picked".
> >
> > ```
> > In the proof of Theorem 2, we are not sure which one the reviewer is pointing to, but we believe should be taken according to algorithm 2.
> > ```
> > In the proof of Theorem 2 in the last revision, the first line of the equation is  $\sum_{t=1}^T \alpha_t^2 = \sum_{t=1}^T \alpha_{g,t}^2(x_{g,t})$. Hence, my question is about $g$ on the RHS. Anyways, this is corrected in the latest revision as $g_t$. So this issue is resolved.

---

> ### Author Response · Authors · 2023-11-23
> **Further Clarification about Corollary 2 by Authors**
>
> We appreciate the further clarification of the previous discussion and constructive comments that help enhance the presentation of our paper. We’d like to add a tentative discussion over the difference from the existing no-regret CBO method in the appendix. In the following, we want to address the reviewer’s remaining concerns.
>
> ```
> The proof of the simple regret in Corollary 2 is "after attaining 13". But the definition $\beta = 1/T$ is established based on the iteration T when 13 is attained. Consequently, "after attaining 13", the upper confidence bound and lower confidence bound (depending on $\beta$) may no longer be applicable. So, this is potentially an error in the proof of the simple regret.
> ```
>
> We understand the concern raised by the reviewer here. Though we leverage the bounds and the acquisition function holds at $ T $ for the $ T+1 $ query, the discrepancy on $\beta$ could be confusing and even erroneous.
>
> We made the following revisions to the corollary 2.
>
> (1) By enlarging $\beta$ from $\beta =2\log(\frac{2(K+1)\vert \tilde{D}\vert T}{\delta})$ to $\beta =2\log(\frac{2(K+1)\vert \tilde{D} \vert (T+1)}{\delta})$, we guarantee that the union bound used in the proof of theorem 1 before attaining 13 would still hold;
>
> (2) we change the acquisition back to $LCB_{f,t}$, since we figured out $UCB_{f,t}$ actually incur an additional term we do not see a straightforward proof to guarantee that it can be bounded by $\alpha_T \leq \epsilon$.
>
> Though we make the changes, the intended interpretation of Corollary 2 remains the same as the following. As long as $T$ is sufficiently large for a fixed $\epsilon$ so that it meets $T \geq \frac{\beta \widehat{\gamma_T} C_1}{\epsilon^2}$, and the algorithm picks the $arg\max_{x \in \tilde{D}\_{\hat{X}\_T} \cap S_{C, T}}LCB_{f,T}(x) $, the algorithm would translate the analysis in theorem 1 into a high probability bound by the simple regret.

---

### Official Review · Reviewer_vk7s · 2023-10-31

**Soundness:** 2 fair
**Presentation:** 3 good
**Contribution:** 3 good
**Rating:** 5
**Confidence:** 3

**Summary:**

This paper proposes a new algorithm for constrained Bayesian optimization. The acquisition function for the objective is a modified UCB, and the acquisition function for the constraints are marginal variance. In each iteration, the algorithm selects a query point by maximizing the maximum over the objective's and constraints' acquisition functions. The authors have proved an upper bound on the number of samples to find an $\epsilon$ confidence interval containing the global maximum. Empirically, they have shown that the proposed method achieves superior convergence speed compared with other constrained Bayesian optimization baselines.

**Strengths:**

- The theoretical result seems to be the first non-asymptotic sample complexity guarantee for constrained Bayesian optimization.
- According to the plots in Figure 3 and Figure 4, it looks like the method is often much better than the baselines when the constraint is more challenging (where the feasible set is a small fraction of the domain).

**Weaknesses:**

- Some important experimental details are missing.
    - It is not clear how to numerically check if $LCB_{f, t, \max}$ is finite or not. To do this, one needs to check if the intersection of $S_{C, t}$ is nonempty, where each super level set $S_{C, t}$ is a non-convex set. It is unclear from the paper how the intersection is computed.
    - It is unclear how the hyperparameter $\beta_t$ is set in the experiments. The constant $\beta_t$ balances the exploration and exploitation and has huge impact on the practical performance.
- The exact condition on the discretization $\tilde D$ is unclear in the statements of Lemma 1 and Theorem 1. According to Lemma 1, $\tilde D$ can be an arbitrary discretization of the domain as long as it contains the global optimum. However, I don't quite think this can be true. For example, the discretization of Srinivas et al. (2009) has to be large enough (exponential) to roughly cover the entire domain. Otherwise, the concentration inequality only holds inside the discretization $\tilde D$ and cannot be generalized to the entire domain.
- The proof relies on an additional assumption that the confidence interval has to shrinks monotonically, which is somewhat non-standard in the analysis of BO algorithms. Though, the author claims they can apply a proof technique by Gotovos et al. (2013) even if this assumption is violated.

**Questions:**

- The Assumption 3 needs $\beta_t$ to be non-increasing. However, subsequently $\beta_t = 2 \log (2 (K + 1) |\tilde D| \pi_t / \delta)$ which is increasing in $t$, given that $\pi_t$ has to grow superlinearly (typically quadratically). Can you comment on this discrepancy?

---

> ### Author Response · Authors · 2023-11-20
> **Response by Authors (1/2)**
>
> We appreciate the detailed comments made by the reviewer. We would like to clarify the confusion raised by the reviewer in the following.
>
> ***Weakness***
>
> ```
> The exact conditions of the discretization are unclear in the statement of lemma 1 and theorem 1.
> ```
>
> We'd like first to clarify the discretization as it is related to the other concern raised by the reviewer. Throughout the design of the algorithm, we focus on dealing with the **finite discretization $\tilde{D}$ of the original search space**, which aligns with many engineering tasks where only the discretization is feasible or a known smoothness on locality alleviates the needs to scan the whole continuous space.
>
> As noted by the reviewer, Srinivas et al. (2009) start by studying a search space of finite discretization and then introducing the smoothness assumption to extend the results to a multi-dimensional continuous search space where the regret bound relies on the search space. We do not further extend the discussion to a continuous search space for two reasons. (1) The motivating application for us does not deal with a continuous search space; (2) There are recent advancements introducing **various treatments** of continuous search space given different assumptions on the structure of the problem. For example, with regard to the high-dimensional BO task as concerned by the reviewer, except directly creating discretization, which incurs the curse of dimensionality, various principled treatments [3,4,5] have been studied. They could be incorporated into our constrained BO framework to address the concern that the construction of a finite search space could be troubled by high dimensionality. A comprehensive discussion of the construction of a finite search space might go beyond the scope of this work.
>
> ```
> It is not clear how to numerically check if $LCB_{f, t, max}$ is finite or not. It is unclear from the paper how the intersection is computed.
> ```
>
> Related to the previous question, we are focusing on a finite discrete search space $\tilde{D}$, where a simple strategy would be checking the membership to different sets using the point-wise LCB and UCB values.
>
> ```
> The constant balances the exploration and exploitation and has a huge impact on the practical performance. It is unclear how it is set in practice.
> ```
>
> We appreciate the constructive comment. We added additional results and corresponding scripts in the revision to demonstrate the robustness against different choices of $\beta$. In summary, among the choices of the theoretical results or constant values, including 0.1, 2, 4, and 8, we observe that only .1 could trap the algorithm in the sub-optimal area due to its lack of exploration.
>
> ```
> The proof relies on an additional assumption that the confidence interval has to shrink monotonically, which is somewhat non-standard in the analysis of BO algorithms. However, the author claims they can apply a proof technique by Gotovos et al. (2013) even if this assumption is violated.
> ```
>
> You are correct that our proof relies on the assumption, and as the reviewer pointed out, we can apply a proof technique by Gotovos et al. (2013) even if this assumption is violated. We can construct a new confidence interval by taking the maximum (minimum) of historical pointwise LCB (UCB). Then, the new confidence interval monotonically and consistently shrinks. Since we are aiming at a guarantee within a finite known $T$ iteration, the construction of this new confidence interval (taking union bound) would only incur a constant cost on the confidence level of the ultimate bound in theorem 1. Please note that such modification implies a slight change in the algorithm. However, we find in our empirical study that, even without these additional steps, our algorithm already exhibits compelling performance consistently across a variety of BO benchmarks.

---

> ### Author Response · Authors · 2023-11-20
> **Response by Authors (2/2)**
>
> ***Question***
>
> ```
> The assumption 2 needs \beta_t to be non-increasing.
> ```
>
> We appreciate the detailed question here. The reason $\pi_t$ needs to grow is the need to guarantee the convergence of the series, which allows an asymptotical sublinear regret. Specifically, the requirement is $\sum_t{\pi_t}^{-1} = 1$. Yet, in our problem setup, since cumulative regret could not be bounded as instantaneous regret could be infinite when the evaluated candidate $x_t$ violates any of the constraints (cf. Equation (3)), we aim to guarantee that after a certain horizon T, the *simple regret* could be bounded. Therefore, there is no need to have a convergence series $\sum_t{\pi_t}^{-1} = 1$. As shown in the proof, we only need $\sum_{t=1}^T{\pi_t}^{-1} = 1$ for a given budget/horizon T, and setting $\pi_t = T$ satisfies it. We made revisions in the analysis section and the proof part to highlight this.
>
> The setting in which $\sum_{t=1}^T{\pi_t}^{-1} = 1$ is used instead of $\sum_t{\pi_t}^{-1} = 1$ can be found in [1] as they also aim at a simple regret after a sufficiently large $T$. Similarly, in a multi-fidelity BO setting [2], the total budget is assumed to be given to attain a theoretical guarantee on the cumulative regret.
>
> ***Reference***
>
> [1] Wang, Zi, and Stefanie Jegelka. "Max-value entropy search for efficient Bayesian optimization." In International Conference on Machine Learning, pp. 3627-3635. PMLR, 2017.
>
> [2] Song, Jialin, Yuxin Chen, and Yisong Yue. "A general framework for multi-fidelity bayesian optimization with gaussian processes." In The 22nd International Conference on Artificial Intelligence and Statistics, pp. 3158-3167. PMLR, 2019.
>
> [3] Maus, Natalie, Haydn Jones, Juston Moore, Matt J. Kusner, John Bradshaw, and Jacob Gardner. "Local latent space Bayesian optimization over structured inputs." Advances in Neural Information Processing Systems 35 (2022): 34505-34518.
>
> [4] Gardner, Jacob, Chuan Guo, Kilian Weinberger, Roman Garnett, and Roger Grosse. "Discovering and exploiting additive structure for Bayesian optimization." In Artificial Intelligence and Statistics, pp. 1311-1319. PMLR, 2017.
>
> [5] Nayebi, Amin, Alexander Munteanu, and Matthias Poloczek. "A framework for Bayesian optimization in embedded subspaces." In International Conference on Machine Learning, pp. 4752-4761. PMLR, 2019.

---

> ### Comment · Reviewer_vk7s · 2023-11-20
>
> Hi authors,
>
> I appreciate the effort of clarification. However, I still disagree the argument on discretization.
>
> First of all, the search domain $\mathbf{X}$ is never assumed to be discrete in the paper. Second, the objective functions used in the experiments have continuous domain (e.g. Rastrigin). If the experiments are conducted on a discretized search space $\tilde D$, the authors are required to provide the detail of the discretization $\tilde D$ (which unfortunately is not available).
>
> This is also related to my original concern: the explicit expression of the discretization $\tilde D$ is missing in Lemma 1. Based on Lemma 1, any discretization works. If I pick a singleton $\tilde D = \\{\mathbf{x}^*\\}$, does the high probability bound still work? It does not quite make sense as the region of interest $\hat{X}_t$ does not make use of the discretization at all..

---

> ### Author Response · Authors · 2023-11-21
> **Response to Concerns about Discretization**
>
> Dear Reviewer vk7s,
>
> Thank you for the swift reply and for clarifying the concerns about the discretization!
>
> ```
> First of all, the search domain $\mathbf{X}$ is never assumed to be discrete in the paper.
> ```
> Before reaching algorithm 1, we do not discuss the discretization $\tilde D$. We define the region of interest $\hat{\mathbf{X}}$ in equation (4) to be a subset of $\mathbf{X}$, which is not assumed to be discrete.
>
> Yet in practice, since we need to check the membership pointwise on the discretization $\tilde D$, we actually rely on the discrete set $\tilde{D}\_{\hat{X_t}}= \tilde D \cap \hat{X_t}$. As illustrated in lines 7-8 of the algorithm 1, we use $arg\max_{x\in \tilde{D}\_{\hat{X_t}} \cap U_{C_k, t}} \alpha_{C_k, t} (x)$ and $arg\max_{x\in \tilde{D}\_{\hat{X_t}}} \alpha_{f, t} (x)$ in contrast to the conceptual discussion in section 4.3 where we use $arg\max_{x\in \hat{X_t} \cap U_{C_k, t}} \alpha_{C_k, t} (x)$ and $arg\max_{x\in \hat{X_t}} \alpha_{f, t} (x)$. We just added additional clarification at the end of section 4.3 to clarify the implementation.
>
>
> ```
> Second, the objective functions used in the experiments have a continuous domain (e.g., Rastrigin). If the experiments are conducted on a discretized search space $\tilde D$, the authors are required to provide the detail of the discretization $\tilde D$ (which, unfortunately, is not available).
> ```
> We just added additional information about the construction of discrete space we used in the low-dimensional synthetic datasets. We fix the $\tilde D$ throughout the experiment on both Ratrigin and Ackley to reduce randomness and guarantee reproducibility. Please note that, as illustrated in the first row of Figure 3, we make the discretization dense to capture the geometry of the continuous functions.
>
>
> ```
> This is also related to my original concern: the explicit expression of the discretization $\tilde D$ is missing in Lemma 1.
> ```
> Yes, you are right! The result of Lemma 1 should actually be the probability bound for $x \in \tilde{D}\_{\hat{X_t}}$ instead of $x \in \hat{X_t}$. We just corrected it in both the lemma and proof in Appendix A.1. We apologize for the confusion. At some point, we decoupled the concepts of search space and ROI from the discretization, which is now separately denoted as $\tilde {D}$, and the notation is mistakenly left over.
>
> -----
> Thank you again for the constructive comments, which we find helpful in enhancing the clarity of the paper! We hope our clarification and updated revision addresses your concern.

---

### Official Review · Reviewer_uw1z · 2023-11-01

**Soundness:** 3 good
**Presentation:** 3 good
**Contribution:** 3 good
**Rating:** 6
**Confidence:** 3

**Summary:**

This paper proposes COBALT, a constrained Bayesian optimization algorithm. It models the unknown objective and constraints with Gaussian processes. Regions of interest (ROIs) are defined for the objective and each constraint based on confidence intervals. Acquisition functions are defined on the ROIs. The next point maximizes the combined acquisition functions over the ROI intersection, enabling an adaptive tradeoff between learning constraints and optimization. Theoretical analysis shows the confidence interval around the global optimum shrinks below a threshold after enough iterations. Experiments demonstrate COBALT efficiently finds the global optimum compared to existing methods, benefitting from active constraint learning. In summary, COBALT introduces a principled constrained Bayesian optimization framework, leveraging ideas from active learning and Bayesian optimization via acquisition functions and ROIs for the adaptive constraint-objective tradeoff.

**Strengths:**

This paper presents an original contribution to the field of constrained Bayesian optimization. The key strength is the principled integration of ideas from both Bayesian optimization and active learning of constraints to develop the novel COBALT algorithm. Defining separate acquisition functions on regions of interest for the objective and constraints enables explicit, adaptive tradeoff between exploring constraints vs exploiting the objective. This combination of existing methods with new problem formulations results in an algorithm with theoretical guarantees on optimizing unknown black box functions subject to unknown constraints. Experiments demonstrate superiority over current state-of-the-art techniques on challenging synthetic and real-world optimization tasks.

**Weaknesses:**

While the COBALT algorithm represents a significant advance, there are a few areas where the work could be strengthened:
* Empirical evaluation is limited to 6 relatively low-dimensional tasks. Testing on more high-dimensional real-world problems would better showcase scalability.
* The theoretical analysis relies on several assumptions that may not always hold in practice, such as the independence of the GPs and the existence of a feasible solution.

**Questions:**

* The theoretical analysis relies on assumptions like GP independence and feasible solutions. Could these be relaxed? What are the barriers to deriving more general guarantees?
* What is the computational complexity of COBALT? How does it scale with dimensionality and constraints?
* Have you tested on any high-dimensional (D>10) or large-scale real-world problems? This could better showcase scalability.
* Is there a principled way to set the β tradeoff parameter? Sensitivity analysis could elucidate its impact.
* How does performance depend on the quality of the GP model of the constraints?

---

> ### Author Response · Authors · 2023-11-20
> **Response by Authors (1/2)**
>
> We are glad the reviewer appreciates our contribution to the BO with unknown constraints, and we sincerely appreciate the constructive comments by the reviewer.
>
> ***Weakness***
> ```
> Empirical evaluation is limited to 6 relatively low-dimensional tasks. Testing on more high-dimensional real-world problems would better showcase scalability.
> ```
>
> 1. Thank you for raising this in the comments. Please note that the water converter is a 36-dimensional task, which is reasonably high-dimensional (i.e., d>10).
>
> 2. As the reviewer noted in the questions, the performance actually depends on the quality of the GP models. We found tuning a large number of GPs in high-dimensional, large-scale tasks to be more challenging. Especially when the prior or kernel learning is not properly tuned, the **misspecification of every single model accumulates in the ROI intersection**. Then, it is possible to result in an empty intersected ROI, which is not expected when models offer satisfying uncertainty quantification.
>
> 3. We do see the potential of further integrating additional ROI-based HDBO treatments, such as BALLET[1], into the COBALT framework to further enhance the scalability through (deep) kernel learning [6]. We want to highlight that, as noted by the reviewer, this submission **focused** on the principled tradeoff between learning and optimization in the constrained BO setting within the limited pages. Advancements in modeling and efficiency for large-scale constrained BO tasks would definitely be of interest in the future.
>
> ```
> The theoretical analysis relies on several assumptions that may not always hold in practice, such as the independence of the GPs and the existence of a feasible solution.
> ```
>
> 1. This is a good point (question). Thank you for raising this concern. For the dependency assumption, we follow the convention in the constrained Bayesian optimization literature [2, 3]. As discussed in section F.3 in the appendix, we acknowledge the limitation of the current assumption and believe it is an interesting future direction to explore different dependency assumptions, which could lead to potentially novel modeling and acquisition function choices.
>
> 2. We also appreciate the reviewer's concern about our assumption of the existence of a feasible solution, although it wasn't a primary consideration in our motivating applications. If this assumption does not hold, it implies two significant implications: (1) the task or application in question may encounter fundamental challenges; (2) as our analysis indicates, the algorithm is designed to identify a subset of the feasible region, should it exist. In the absence of such a region, the active learning component of our algorithm would converge to an empty intersected Region of Interest (ROI).
>
> 3. To provide further clarity on this aspect, we have extended our analysis in the appendix of the revised version. Our additional result further highlights the algorithm's capability to detect infeasibility and is presented as a corollary of our existing results.

---

> > ### Author Response · Authors · 2023-11-20
> > **Response by Authors (2/2)**
> >
> > ***Questions***
> >
> > ```
> > The theoretical analysis relies on assumptions like GP independence and feasible solutions. Could these be relaxed? What are the barriers to deriving more general guarantees?
> > ```
> > Answered in the weakness section.
> >
> > ```
> > What is the computational complexity of COBALT? How does it scale with dimensionality and constraints?
> > ```
> >
> > Depending on the modeling choice, the computational complexity of COBALT could vary. Specifically, if the conventional BO computational complexity is $A$, the number of unknown constraints is $K$, then the computational complexity of  COBALT would be $\mathcal{O}((K+1)A)$, as it essentially duplicates the learning and inference procedure, which dominates the complexity of the single-GP BO. By default, the conventional GP incurs an $\mathcal{O}(N^3)$ cost in the BO procedure, where N is the number of historical observations. However, there are recent advancements in applying kernel interpolation [4] or replacing GP with other efficient surrogate models for BO[5]. Incorporating these techniques is definitely of interest to a more scalable COBALT variant.
> >
> > We also report additional wall-time comparisons in the revised appendix.
> >
> > ```
> > Have you tested on any high-dimensional (D>10) or large-scale real-world problems? This could better showcase scalability.
> > ```
> > Answered in the weakness section.
> >
> > ```
> > Is there a principled way to set the β tradeoff parameter? Sensitivity analysis could elucidate its impact.
> > ```
> >
> > We appreciate the constructive comment and add a brief study of the sensitivity of $\beta$ in Section E in the revision. In summary, among the choices of the theoretical results or constant values, including 0.1, 2, 4, and 8, we observe that only .1 could trap the algorithm in the sub-optimal area due to its lack of exploration. We acknowledge that there is a discrepancy in the theory and practice of $\beta$ as in the literature. In the case of COBALT, since the estimation of $\epsilon$ plays an important role while might not always be available in prior, we typically take a moderate value, as studied above, which shows robust performance.
> >
> > ```
> > How does performance depend on the quality of the GP model of the constraints?
> > ```
> > Answered in the weakness section.
> >
> >
> > ***Reference***
> >
> > [1] Zhang, Fengxue, Jialin Song, James C. Bowden, Alexander Ladd, Yisong Yue, Thomas Desautels, and Yuxin Chen. "Learning Regions of Interest for Bayesian Optimization with Adaptive Level-Set Estimation." (2023).
> >
> > [2] Takeno, Shion, Tomoyuki Tamura, Kazuki Shitara, and Masayuki Karasuyama. "Sequential and parallel constrained max-value entropy search via information lower bound." In International Conference on Machine Learning, pp. 20960-20986. PMLR, 2022.
> >
> > [3] Gelbart, Michael A., Jasper Snoek, and Ryan P. Adams. "Bayesian optimization with unknown constraints." arXiv preprint arXiv:1403.5607 (2014).
> >
> > [4] Wilson, Andrew, and Hannes Nickisch. "Kernel interpolation for scalable structured Gaussian processes (KISS-GP)." In International conference on machine learning, pp. 1775-1784. PMLR, 2015.
> >
> > [5] Shahriari, Bobak, Kevin Swersky, Ziyu Wang, Ryan P. Adams, and Nando De Freitas. "Taking the human out of the loop: A review of Bayesian optimization." Proceedings of the IEEE 104, no. 1 (2015): 148-175.
> >
> > [6] Wilson, Andrew Gordon, et al. "Deep kernel learning." Artificial intelligence and statistics. PMLR, 2016.

---

### Author Response · Authors · 2023-11-20
**General Response by Authors**

We sincerely appreciate the detailed and constructive comments by reviewers. We offered responses and clarification to the concerns and confusions raised by all the reviewers and submitted the revised paper. We highlighted the major changes in the revised paper, including but not limited to the additional discussion over the literature brought up by the reviewers, the corrected description of the algorithm, the updated theoretical results in the main content, and additional results in the appendix. Once again, we want to thank all the reviewers for their insightful comments. We highly appreciate your questions and hope our responses enhance your perception of our paper.

---

> ### Author Response · Authors · 2023-11-23
>
> Dear Reviewers,
>
> We sincerely appreciate your active engagement and valuable feedback. The ongoing discussion has been invaluable, especially during this concluding phase of the author-reviewer discussion. Despite the limited time for further deliberation, we want to express our openness to providing any additional clarifications necessary to thoroughly address the concerns raised by all reviewers.
>
> Thank you for your time and consideration.

---

### Meta-Review · Area_Chair_C8Q4 · 2023-12-07

**Metareview:**

The authors consider a constrained Bayesian optimization setting where the constraints may not be known a priori but only learned during the optimization process. The authors propose a algorithm for this setting, analyze it theoretically, and evaluate it empirically.

The reviewers generally agree that the problem setting is of interest to the ICLR audience and that the authors' overall approach is reasonable. The reviewers in particular commended the authors for their theoretical analysis on constrained Bayesian optimization, which is not common to see in this particular subarea.

However, the reviewers also noted some perceived weaknesses with the manuscript as submitted:

- a series of non-trivial questions regarding the theoretical analysis, with some reviewers expressing a lack of confidence in claims as a result
- a lack of clarity regarding connections to existing work and, as a result, the novelty of the contributions here

**Justification For Why Not Higher Score:**

Ultimately the perceived weaknesses identified by the reviewers were deemed to serious to overcome, or to be addressed in minor revisions, despite extensive engagement with the authors during the author response period.

**Justification For Why Not Lower Score:**

N/A

---

### Decision · Program_Chairs · 2024-01-16

Reject